



# Wind turbine wake simulation with explicit algebraic Reynolds stress modeling

Mads Baungaard[1], Stefan Wallin[2], Maarten Paul van der Laan[1], and Mark Kelly[1]

[1]DTU Wind Energy, Technical University of Denmark, Risø Campus, Frederiksborgvej 399, 4000 Roskilde, Denmark
[2]Department of Mechanics, Linné FLOW centre, KTH Royal Institute of Technology, Osquars Backe 18, 10044 Stockholm, Sweden

**Correspondence:** Mads Baungaard (mchba@dtu.dk)

**Abstract.** Reynolds-averaged Navier-Stokes (RANS) simulations of wind turbine wakes are usually conducted with two-equation turbulence models based on the Boussinesq hypothesis, which are simple and robust but lack the capability of predicting various turbulence phenomena. Using the explicit algebraic Reynolds stress model (EARSM) of Wallin and Johansson (2000) can aid some of these deficiencies, while still being numerical robust and only slightly more computationally expensive than the traditional two-equation models. The model implementation is verified with the homogeneous shear flow, half-channel flow and square duct flow cases, and subsequently full 3D wake simulations are run and analyzed. The results are compared with reference large eddy simulation (LES) data, which shows that the EARSM especially improves the prediction of turbulence anisotropy and turbulence intensity but that it also predicts less Gaussian shaped wake profiles with the standard settings of the model.

## 1 Introduction

As wind farms increase in size and number of turbines, increasingly more attention should be given to the study of wind turbine wakes as they can account for a relatively large power production decrease. Simple engineering models, e.g., the classic Jensen (1983) model, the "new-classic" Bastankhah and Porté-Agel (2014) model or the more recent Ishiara model (Ishihara and Qian, 2018), can be used to model the flow through a wind farm on a regular laptop in a matter of seconds. However, they all have in common that they are based on rather strict flow assumptions, need empirical tuning parameters and a superposition model for overlapping wakes.

The wind turbine wake is complex since it is a three-dimensional, swirling flow and its development is governed by turbulence mixing, which is strongly influenced by density stratification in the atmospheric surface layer (ASL) and the interaction between the ASL and the turbulent wake. To model a more physical correct wind farm flow we therefore solve the set of Navier-Stokes equations, which in essence is a reformulation of Newton's second law and conservation of mass. The process of discretizing and solving these equations on computers is known as computational fluid dynamics (CFD). Unfortunately, the Reynolds number of atmospheric flows is so large, that it is unfeasible to conduct direct numerical simulations (DNSs), even for a single wind turbine wake. Instead large eddy simulation (LES), i.e., the simulation of the spatially-filtered Navier-Stokes equations, have been performed by several groups in the last decade, see review by Breton et al. (2017). Even though the rotor





geometry is reduced to an actuator disk (AD) or an actuator line (AL) model, these simulations require a vast amount of CPU hours on modern high-performance computing (HPC) clusters, again even for a single wind turbine wake. A much faster simulation can be conducted by solving the Reynolds-averaged Navier-Stokes (RANS) equations instead, but this type of simulation relies heavily on the quality of the turbulence model because all turbulence scales are modeled. For example, the standard two-equation $k$–$\varepsilon$ model of Launder and Spalding (1974) was shown by Réthoré (2009) to perform poorly for simulation of wind

turbine wakes in terms of wake recovery prediction and having unphysical Reynolds stresses in the vicinity of the turbine, i.e., being unrealizable. This is not unexpected since it was originally developed for simple free shear and boundary-layer flows, e.g., flat plate, pipe, plane jet and cavity flows (Launder and Spalding, 1974). The standard $k$–$\varepsilon$ model is considered to be a linear eddy viscosity model (EVM), because the anisotropy tensor, $a_{ij} \equiv \frac{\overline{u'_i u'_j}}{k} - \frac{2}{3}\delta_{ij}$ ($\overline{u'_i u'_j}$ is the Reynolds stress tensor, $k$ is the turbulent kinetic energy (TKE) and $\delta_{ij}$ is the Kronecker delta tensor), is linearly proportional to the normalized strain

rate tensor through the Boussinesq hypothesis (Boussinesq, 1897), see also Eq. (5). A modified model coined the "$k$–$\varepsilon$–$f_P$" model was developed and calibrated specifically for atmospheric wind farm flows by van der Laan (2014) and it showed much improvement over the standard $k$–$\varepsilon$ model. It is essentially equivalent to the first order model of the non-linear eddy viscosity model (NLEVM) of Apsley and Leschziner (1998).

   Linear EVMs based on the Boussinesq hypothesis, e.g., mixing-length, $k$–$\varepsilon$, $k$–$\omega$, etc., are the de-facto standard for turbu-

lence modeling in many fields of research including wind energy applications. They are simple to implement and numerical robust but have a drawback in terms of physical correctness - the example of wake recovery has already been mentioned earlier and one can also quickly derive that the Reynolds stresses become unrealizable for large normalized strain rates in general, see Fig. 2. Full Reynolds stress modeling (RSM), where a transport equation for each of the six unique Reynolds stress components is solved, was conducted by Launder et al. (1975) and this approach leads to more physical predictions, because one avoids the

use of the quite limiting Boussinesq hypothesis, and hence the production of turbulence needs no further modelling, which is a major improvement. The drawback of this method is that it is more computationally expensive and, perhaps more importantly, less numerically stable. Rodi (1976) deployed the weak equilibrium approximation (WEA) to the RSM equations, which transforms the set of differential equations into a set of algebraic equations, while still retaining most of the physical behavior of the RSM. This approach is known as the algebraic Reynolds stress model (ARSM) and it can be formulated as a tensor equation

for the anisotropy tensor, but unfortunately it is a non-linear implicit equation with multiple solutions, which is also prone to numerical stability issues. Pope (1975) simplified the ARSM by using the Cayley-Hamilton theorem and obtained the first explicit algebraic Reynolds stress model (EARSM) for the case of 2D flow, which as the name suggests is an explicit, algebraic relation for the Reynolds stress tensor (or equivalently the anisotropy tensor) given the normalized strain rate and rotation rate tensors. Different generalizations to 3D were made by Taulbee (1992), Gatski and Speziale (1993), and Wallin and Johansson

(2000), which differ in the constants and more importantly in the way they treat the non-linearity of the ARSM. The Wallin and Johansson (2000) EARSM (WJ-EARSM) is preferred, because it is based on the concept of self-consistency (meaning that the explicit solution of $a_{ij}$ satisfies the non-linear ARSM tensor equation exactly in 2D mean flows and approximately in 3D mean flows) and is only slightly more computationally expensive compared to a standard $k$–$\varepsilon$ model. The self-consistent solution





was independently formulated by Girimaji (1996), Ying and Canuto (1996) and Johansson and Wallin (1996), and facilitates a
consistent solution in non-equilibrium conditions.

It appears that only Gómez-Elvira et al. (2005) and van der Laan (2014) have attempted using EARSM for RANS simulations
of wind turbine wakes. Many other research areas have however used the WJ-EARSM successfully (e.g., airfoil flow (Franke
et al., 2005), vortex generators (Jirásek, 2005), stirred tanks (Feng et al., 2012), Kaplan turbines (Javadi and Nilsson, 2017) and
high-speed trains (Munoz-Paniagua et al., 2017)), so it seems that there exist an untapped potential in turbulence modeling for
wind energy applications. Gómez-Elvira et al. (2005) deployed the Taulbee (1992) model in a parabolic RANS setup, which
is fast to execute but lacks physical features, e.g., the upstream induction zone. van der Laan (2014) tested the Taulbee (1992),
Gatski and Speziale (1993) and Apsley and Leschziner (1998) models in the elliptic RANS solver EllipSys3D (Sørensen, 1995)
but found that they were more numerically unstable compared to the standard $k$–$\varepsilon$ model.

The EARSM framework of Wallin and Johansson (2000) has been further exploited by including the strong coupling with
density stratification present in the ASL (Lazeroms et al., 2013) capturing the effect of both stable and convective atmospheric
boundary layers (ABLs) (Lazeroms et al., 2016; Želi et al., 2020, 2021). In this paper, we will restrict the study to neutral ASLs
leaving stratified conditions for upcoming studies.

Section 2 describes the turbulence model formulations. In Section 3, we verify our implementation of the WJ-EARSM using
several canonical flowcases and finally in Sections 4 and 5, it is applied to simulations of wind turbine wakes in the neutral
ASL.

## 2 Turbulence modeling

The turbulence models utilized in this paper assume incompressible, non-stratified flow, no system rotation (no Coriolis or
centrifugal contributions), isotropic dissipation of TKE and high Reynolds number flow.

### 2.1 The standard $k$–$\varepsilon$ model (Launder and Sharma, 1974)

The Boussinesq hypothesis is used to obtain the Reynolds stresses

$$\overline{u'_i u'_j} = -\nu_t \left( \frac{\partial U_i}{\partial x_j} + \frac{\partial U_j}{\partial x_i} \right) + \frac{2}{3} k \delta_{ij}, \tag{1}$$

which are needed to close the momentum equations (the equations for the mean velocity vector, $U_i$). The eddy viscosity in a
linear EVM is defined as $\nu_t = C_\mu \frac{k^2}{\varepsilon}$ and transport equations are used to obtain TKE, $k$, and dissipation of TKE, $\varepsilon$:

$$\frac{\partial k}{\partial t} + U_j \frac{\partial k}{\partial x_j} = \underbrace{-\overline{u'_i u'_j} \frac{\partial U_i}{\partial x_j}}_{\mathcal{P}} - \varepsilon + \underbrace{\frac{\partial}{\partial x_j} \left( \frac{\nu_t}{\sigma_k} \frac{\partial k}{\partial x_j} \right)}_{\mathcal{D}^{(k)}}, \tag{2}$$

$$\frac{\partial \varepsilon}{\partial t} + U_j \frac{\partial \varepsilon}{\partial x_j} = (C_{\varepsilon 1} \mathcal{P} - C_{\varepsilon 2} \varepsilon) \frac{\varepsilon}{k} + \underbrace{\frac{\partial}{\partial x_j} \left( \frac{\nu_t}{\sigma_\varepsilon} \frac{\partial \varepsilon}{\partial x_j} \right)}_{\mathcal{D}^{(\varepsilon)}}. \tag{3}$$





This paper focuses on the $k$–$\varepsilon$ model, because it is usually preferred for atmospheric flows, while $k$–$\omega$ models are more popular for engineering flows. They can both be categorised as linear EVMs, because they use the Boussinesq hypothesis to obtain the Reynolds stresses. Several empirical constants are also present for both models; for the $k$–$\varepsilon$ model there are $C_\mu$, $\sigma_k$, $C_{\varepsilon 1}$, $C_{\varepsilon 2}$ and $\sigma_\varepsilon$ (see review by Weaver and Mišković (2021) for the most popular sets of constants used in the past). We shall use

different sets of constants throughout the paper and will remark the choice at each usage of the $k$–$\varepsilon$ model.

To simplify the Boussinesq hypothesis, Eq. (1), we can use the anisotropy tensor also mentioned in the introduction:

$$\mathbf{a} = a_{ij} \equiv \frac{\overline{u_i' u_j'}}{k} - \frac{2}{3}\delta_{ij}. \tag{4}$$

It is dimensionless, symmetric and traceless, and Eq. (1) can then be re-written as:

$$a_{ij} = -2C_\mu S_{ij} \quad \text{(linear EVM)}, \tag{5}$$

where $\mathbf{S} = S_{ij} \equiv \frac{1}{2}\frac{k}{\varepsilon}\left(\frac{\partial U_i}{\partial x_j} + \frac{\partial U_j}{\partial x_i}\right)$ is the normalized strain rate tensor. The $k$–$\varepsilon$ model is independent of the normalized rotation rate tensor, $\mathbf{\Omega} = \Omega_{ij} \equiv \frac{1}{2}\frac{k}{\varepsilon}\left(\frac{\partial U_i}{\partial x_j} - \frac{\partial U_j}{\partial x_i}\right)$, hence it will be unable to predict turbulence effects associated with rotation or curvature, e.g., damping/enhancement of turbulence in rotating homogeneous shear flow (Wallin and Johansson, 2002). The more advanced EARSMs to be discussed next, also need to solve the $k$ and $\varepsilon$ transport equations, because they depend on $S_{ij}$ and $\Omega_{ij}$. The key difference between these more advanced models and the standard $k$–$\varepsilon$ closure is that the Boussinesq

hypothesis is replaced by a more general constitutive relation.

Finally, it can be noted that the time derivative is retained in the transport equations, Eq. (2)-(3), to allow for unsteady RANS (URANS) simulations, e.g., homogeneous shear flow. URANS is generally only advisable, when there is a clear separation between the time scale of the turbulence and the mean unsteadiness, see for example Wallin (2000), and should therefore be used with care.

**2.2 The $k$–$\varepsilon$–$f_P$ model**

The $k$–$\varepsilon$–$f_P$ model by van der Laan (2014) is equivalent to the first order model of Apsley and Leschziner (1998), except for a re-tuning of the model coefficients. To summarize:

$$a_{ij} = -2\underbrace{C_\mu f_P}_{C_\mu^{\text{eff}}} S_{ij} \quad (k\text{-}\varepsilon\text{-}f_P \text{ model}). \tag{6}$$

Compared with the Boussisnesq hypothesis, Eq. (5), we see that the only difference is that the $k$–$\varepsilon$–$f_P$ model uses a variable

or "effective" $C_\mu^{\text{eff}}$, which is flow dependent and thereby makes $a_{ij}$ non-linear in the velocity gradient tensor; however, the $k$–$\varepsilon$–$f_P$ model is still referred to as a "linear EVM", e.g. van der Laan (2014), because the direction of $a_{ij}$ is aligned with $S_{ij}$ and not any other higher order tensors. The $f_P$ function is:

$$f_P = \frac{2f_0}{1 + \sqrt{1 + 4f_0(f_0 - 1)(\sigma/\tilde{\sigma})^2}} \quad , \quad \sigma \equiv \frac{k}{\varepsilon}\sqrt{\left(\frac{\partial U_i}{\partial x_j}\right)^2} = \sqrt{II_S - II_\Omega} \quad , \quad f_0 = 1 + \frac{1}{C_R - 1}. \tag{7}$$





In the above equation $\sigma$ is the "shear parameter", which can re-written using velocity gradient tensor invariants, see Eq. (14), while $\tilde{\sigma}$ is the shear parameter of the freestream calibration flow. Instead of using DNS channel flow data for the calibration, as was done by Apsley and Leschziner (1998), van der Laan (2014) chose to calibrate with a neutral ASL, which simply gives $\tilde{\sigma} = C_\mu^{-1/2}$. Finally, van der Laan (2014) took the Rotta constant as a free parameter and tuned it to $C_R = 4.5$ using LES data of wake velocity and TI (pressure-strain data was not used) extracted from eight different wind turbine wake cases.

In the freestream of a neutral ASL, $f_P = 1$, so it reduces to the standard $k$–$\varepsilon$ model, while $f_P < 1$ in regions with rapid strain compared to the turbulence time scale (i.e., large normalized velocity gradients), hence attenuating mixing in the wake shear layers and improving predictions of wind turbine wakes as found by van der Laan (2014). The $f_P$ correction reduces mainly the turbulence length scale (but also the turbulence velocity scale) in the near wake, and can therefore be interpreted as a local turbulence length scale limiter, as discussed in van der Laan and Andersen (2018). In fact, the damping introduced through $f_P$ will preserve realizability for rapid shear, e.g. in the RDT limit $a_{12} > -1$, while the standard $k$–$\varepsilon$ model will become unrealizable, see Fig. 2.

Table 1 summarizes the model constants of the $k$–$\varepsilon$–$f_P$ model, where one can notice that $C_\mu = 0.03$ is used instead of the established value of $C_\mu = 0.09$, e.g., Launder and Spalding (1974). Several measurements of flat terrain, atmospheric flows, e.g., Panofsky and Dutton (1984), points to a lower $C_\mu$ and Bottema (1997) argues that this is due to "inactive" low frequency atmospheric turbulence, see also discussion by Richards and Norris (2011).

| $C_{\varepsilon,1}$ | $C_{\varepsilon,2}$ | $\sigma_k$ | $\sigma_\varepsilon$ | $C_\mu$ | $\kappa$ | $C_R$ |
|---|---|---|---|---|---|---|
| 1.21 | 1.92 | 1.00 | 1.30 | 0.03 | 0.4 | 4.5 |

**Table 1.** Model constants for the $k$–$\varepsilon$–$f_P$ model as recommended by Sørensen (1995) and van der Laan (2014). Referred to later as "ABL coefs".

## 2.3 Wallin and Johansson (2000) EARSM

The EARS model of Wallin and Johansson (2000) is derived from the ARSM of Rodi (1976) and therefore inherits the constants, $c_1$ and $c_2$, which are the Rotta coefficient and rapid pressure-strain coefficient of the Launder et al. (1975) model, respectively. The particular choice $c_2 = 5/9$ reduces the model expressions significantly and will be adopted in this study. This choice is also supported by the DNSs of Shabbir and Shih (1993). Moreover, we will only consider the incompressible, high-Re version (without near-wall corrections) due to the high Reynolds number and the use of rough wall function boundary conditions at the ground in the considered flow cases of this paper. Additionally, we will make comparisons between the 2D and 3D models. It is important to stress that the 2D model is fully general and invariant and can be used for simulation of 3D mean flows as noted by Hellsten and Wallin (2009); this has for example also been commonly done with the EARSM of Gatski and Speziale (1993). Another important point to mention again is that all EARSMs are based on combinations of $S_{ij}$ and $\Omega_{ij}$ (which depend on the turbulence time scale, $\tau = k/\varepsilon$) so one still needs to solve either the $k$–$\varepsilon$ model (used in this paper), $k$–$\omega$ model (used by Wallin and Johansson (2000)) or some other combination for obtaining the turbulence time scale.



### 2.3.1 2D WJ-EARSM

Using the complete two-dimensional tensor basis for the anisotropy tensor (Pope, 1975), an exact self-consistent 2D EARSM
was found independently by Johansson and Wallin (1996), Girimaji (1996) and Ying and Canuto (1996). It was more thor-
oughly elaborated and tested by Wallin and Johansson (2000), where derivation details also can be found. Without loss of
generality, and for the reason of numerical implementation, the anisotropy is split into linear and "extra" terms:

$$a_{ij} = -2C_\mu^{\text{eff}} S_{ij} + a_{ij}^{(\text{ex})}, \tag{8}$$

where

$$C_\mu^{\text{eff}} = -\frac{1}{2}\beta_1 \quad , \quad a_{ij}^{(\text{ex})} = \beta_4 \left( S_{ik}\Omega_{kj} - \Omega_{ik}S_{kj} \right). \tag{9}$$

The tensor coefficients are:

$$\beta_1 = -\frac{6}{5}\frac{N}{N^2 - 2II_\Omega}, \tag{10}$$

$$\beta_4 = -\frac{6}{5}\frac{1}{N^2 - 2II_\Omega}, \tag{11}$$

while $N$ is the real and positive root of a cubic polynomial related to the non-linearity of the EARSM:

$$N = \begin{cases} \frac{c_1'}{3} + \left(P_1 + \sqrt{P_2}\right)^{1/3} + \text{sign}\left(P_1 - \sqrt{P_2}\right) |P_1 - \sqrt{P_2}|^{1/3}, & P_2 \geq 0 \\ \frac{c_1'}{3} + 2\left(P_1^2 - P_2\right)^{1/6} + \cos\left(\frac{1}{3}\left(\frac{P_1}{\sqrt{P_1^2 - P_2}}\right)\right), & P_2 < 0 \end{cases} \tag{12}$$

$$P_1 = \left(\frac{1}{27}c_1'^2 + \frac{9}{20}II_S - \frac{2}{3}II_\Omega\right)c_1' \quad , \quad P_2 = P_1^2 - \left(\frac{1}{9}c_1'^2 + \frac{9}{10}II_S + \frac{2}{3}II_\Omega\right)^3 \quad , \quad c_1' = \frac{9}{4}(c_1 - 1). \tag{13}$$

The model only depends on the first two velocity gradient tensor invariants:

$$II_S \equiv S_{ij}S_{ji}, \quad II_\Omega \equiv \Omega_{ij}\Omega_{ji}. \tag{14}$$

The solution procedure will be more thoroughly described in Sect. 2.4, but one can just notice that given $S_{ij}$ and $\Omega_{ij}$, there
is a closed and explicit solution of $a_{ij}$. Since $N$, Eq. (12), is an exact solution of the underlying cubic polynomial problem,
then $a_{ij}$ is an exact solution of the ARSM, hence it is a self-consistent EARSM, which ensures good model predictions in non-
equilibrium conditions. We suspect that the treatment of the EARSM non-linearity by Taulbee (1992); Gatski and Speziale
(1993); Apsley and Leschziner (1998) is one of the main causes for the numerical instability when these EARSMs/NLEVMs
are employed to model wind turbine wakes (van der Laan, 2014).

| $C_{\varepsilon,1}$ | $C_{\varepsilon,2}$ | $\sigma_k$ | $\sigma_\varepsilon$ | $\kappa$ | $c_1$ | $c_2$ |
|---|---|---|---|---|---|---|
| 1.44 | 1.82 | 1.00 | 1.30 | 0.38 | 1.8 | $\frac{5}{9}$ |

**Table 2.** Model constants for the WJ-EARSM as recommended by Želi et al. (2020). Referred to later as "Zeli coefs".



$C_\mu$ is not an input parameter but output as a result of applying the WJ-EARSM, see Eq. (9). However, one still needs it

for the rough wall boundary condition (BC) used in our code and for the diffusion terms in the $k$ and $\varepsilon$ transport equations in our implementation; here we shall use $C_\mu = 0.087$, which is the equilibrium value (the value obtained by fixing $\mathcal{P}/\varepsilon = 1$ and considering the log-layer relations) with the current set of constants in Table 2, see Sect. 4.1. Also note that $\kappa = 0.38$ is not an input parameter, but is calculated from the other model constants in order to satisfy the log-layer balance, see Eq. (23).

### 2.3.2   3D WJ-EARSM

The 3D model is derived in an analogous way as the 2D model (Wallin and Johansson, 2000) but with the complete three-dimensional tensor representation of $a_{ij}$, which gives:

$$a_{ij} = -2C_\mu^{\mathrm{eff}} S_{ij} + a_{ij}^{(\mathrm{ex})}, \tag{15}$$

with

$$C_\mu^{\mathrm{eff}} = -\frac{1}{2}(\beta_1 + II_\Omega \beta_6) \quad , \quad a_{ij}^{(\mathrm{ex})} = \beta_3 T_3 + \beta_4 T_4 + \beta_6 (T_6 - II_\Omega S_{ij}) + \beta_9 T_9. \tag{16}$$

The tensor coefficients and basis tensors are:

$$
\begin{aligned}
\beta_1 &= -\frac{N(2N^2 - 7II_\Omega)}{Q} & & \\
\beta_3 &= -\frac{12N^{-1}IV}{Q} & T_3 &= \Omega^2 - \frac{1}{3}II_\Omega \mathbf{I} \\
\beta_4 &= -\frac{2(N^2 - 2II_\Omega)}{Q} & T_4 &= S\Omega - \Omega S \\
\beta_6 &= -\frac{6N}{Q} & T_6 &= S\Omega^2 + \Omega^2 S - \frac{2}{3}IV\mathbf{I} \\
\beta_9 &= \frac{6}{Q} & T_9 &= \Omega S\Omega^2 - \Omega^2 S\Omega
\end{aligned}
,
$$

where

$$Q = \frac{5}{6}(N^2 - 2II_\Omega)(2N^2 - II_\Omega), \tag{17}$$

$$IV = S_{ij}\Omega_{jk}\Omega_{ki}. \tag{18}$$

Unfortunately, there does not exist an analytical solution for $N$ in the 3D model; therefore the cubic $N$ solution from 2D, Eq. (12), is used, hence the 3D model is not exactly self-consistent, but the cubic $N$ solution is however still a quite good approximation in many cases. The same model constants are also used, see Table 2.

One can show that the 3D model reduces to the 2D model in 2D coplanar mean flows, where $T_6 - II_\Omega S = 0$, $IV = 0$ and $T_9 = -\frac{1}{2}II_\Omega T_4$ are valid.

### 2.4   Implementation details

The flow cases are simulated with EllipSys3D, which is a finite-volume CFD solver developed and described in detail by Michelsen (1992) and Sørensen (1995). The solver already has implementations of the standard $k$–$\varepsilon$ model and the $k$–$\varepsilon$–$f_P$





model (van der Laan, 2014), so the following focuses on the implementation of the WJ-EARSM. As emphasized in the previous sections, both the 2D and 3D models can be written in the same form, see Eq. 8 and 15 (although with different expressions

for $C_\mu^{\text{eff}}$ and $a_{ij}^{(ex)}$). The splitting of the anisotropy tensor into a linear and an extra part makes the implementation relatively straightforward in codes that already have a $k$–$\varepsilon$ model implemented as also noted in Appendix A of Wallin and Johansson (2000); for the momentum equations, simply use $\nu_t^{\text{eff}} = C_\mu^{\text{eff}} \frac{k^2}{\varepsilon}$ instead of $\nu_t = C_\mu \frac{k^2}{\varepsilon}$ and add $-\frac{\partial a_{ij}^{(ex)} k}{\partial x_j}$ as a source term:

$$-\frac{\partial \overline{u'_i u'_j}}{\partial x_j} = -\frac{\partial \left(a_{ij} k + \frac{2}{3} k \delta_{ij}\right)}{\partial x_j} = \underbrace{\frac{\partial 2 C_\mu^{\text{eff}} k S_{ij}}{\partial x_j}}_{\text{Treat implicit}} - \underbrace{\frac{\partial a_{ij}^{(ex)} k}{\partial x_j}}_{\text{Treat explicit}} - \underbrace{\frac{\partial \frac{2}{3} k \delta_{ij}}{\partial x_j}}_{\text{Absorb into pressure}} . \tag{19}$$

For numerical stability, it is recommended to include the first term in the system matrix and the second term in the source

vector (i.e., treat the terms implicit and explicit, respectively). The third term is isotropic and can be absorbed into a modified pressure.

In the $k$ and $\varepsilon$ transport equations, we use the standard $\nu_t$ in the diffusion terms ($\mathcal{D}^{(k)}$ and $\mathcal{D}^{(\varepsilon)}$ in Eq. (2)-(3)), since these are calibrated using the standard model. This practice is also used by Apsley and Leschziner (1998), Myllerup (2000) and Menter et al. (2009). In the original paper of Wallin and Johansson (2000), it was proposed to use the Daly-Harlow diffusion model,

or an eddy-diffusion model with the effective $\nu_t^{\text{eff}}$, which was later abounded for the standard eddy-diffusivity model, see e.g. Menter et al. (2012). The TKE production is calculated consistently as $\mathcal{P} = -\varepsilon a_{ik} S_{ki}$ with the full anisotropy tensor.

A segregated solver (i.e., solving $U$-mom'm eq., then $V$-mom'm eq., then $W$-mom'm eq., etc.) is used in EllipSys and the same $\mathcal{P}$ is used in both turbulence transport equations; an overview of the procedure is sketched in Fig. 1.



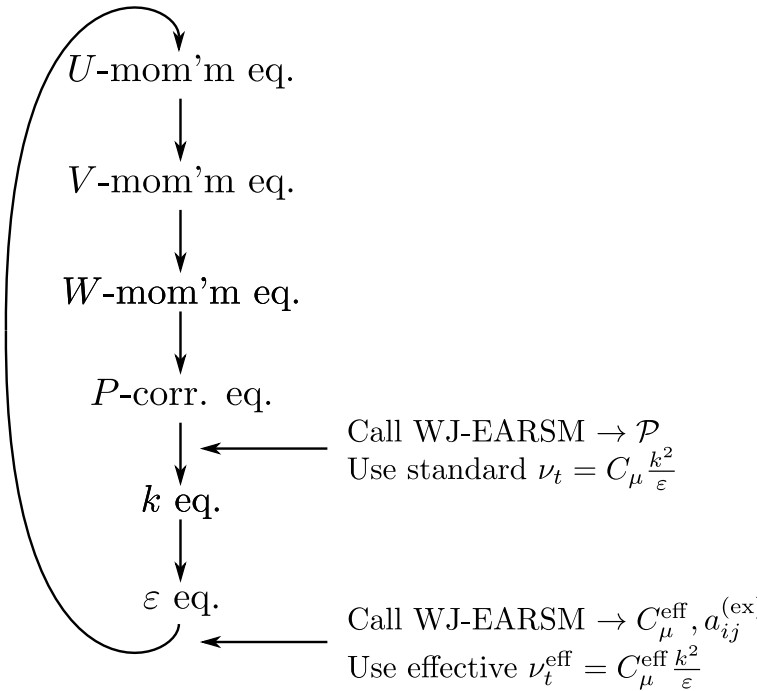

**Figure 1.** Segregated solver procedure for WJ-EARSM in EllipSys3D. Note, $P$ is the mean pressure, while $\mathcal{P}$ is the TKE shear production.

The 1D version of EllipSys3D, EllipSys1D (van der Laan and Sørensen, 2017), is used for the 1D verification cases in
Section 3. Its code structure is very similar to EllipSys3D's, but it has no $W$-momentum equation ($W = 0$), no pressure
correction equation ($\partial/\partial x = \partial/\partial y = 0$, so continuity is ensured already) and it is written using 1D simplifications (mean
variations only in time and z-direction), which makes it possible to run 1D simulations, or so called "single-column models",
on a regular laptop in a few seconds.

For both EllipSys1D and EllipSys3D, the procedure of calling the WJ-EARSM is as such:

1. Use most recent solution of momentum and turbulence transport equations to calculate the normalized strain rate and
    rotation rate tensors, $S_{ij}$ and $\Omega_{ij}$.

2. Calculate tensors and invariants.

3. Calculate $N$.

4. Calculate coefficients, $\beta_1, \ldots, \beta_{10}$.

5. Calculate anisotropy tensor, $a_{ij}$.

6. Calculate TKE shear production, $\mathcal{P} = -\varepsilon a_{ik} S_{ki}$.

7. Calculate $C_\mu^{\text{eff}}$ and $a_{ij}^{(\text{ex})}$.





## 3 Verification cases

As is clear from the previous section, the expressions in the WJ-EARSM are considerably longer compared to the ones of the $k$–$\varepsilon$ and $k$–$\varepsilon$–$f_P$ models. Three canonical flows (homogeneous shear flow, half-channel flow and square duct flow) are therefore used as verification cases to gain confidence in the numerical implementation; full 3D wind turbine wake simulations will first be considered in Sect. 4. The first two cases can be simulated in a 1D setup, which make them ideal for initial testing, while the last case needs to be simulated in 3D due to the phenomenon of secondary motions. A quasi-2D setup - homogeneous in the streamwise direction - would be possible, but a fully 3D setup is chosen for verification of the 3D implementation. All cases are compared to either analytic expressions or DNS data to verify correct behavior of the implementation.

Even before running the verification cases, we consider the general class of "simple shear flows" (aka. 1D parallel flows), where the normalized strain rate and rotation rate tensors are:

$$
S_{simple} = \begin{pmatrix} 0 & 0 & S_{13} \\ 0 & 0 & 0 \\ S_{13} & 0 & 0 \end{pmatrix} \quad , \quad \Omega_{simple} = \begin{pmatrix} 0 & 0 & S_{13} \\ 0 & 0 & 0 \\ -S_{13} & 0 & 0 \end{pmatrix}. \tag{20}
$$

Prescribing $S_{13}$ enables us to evaluate the turbulence models analytically and obtain the anisotropy tensor, $a_{ij}$. The $a_{13}$ component is shown in Fig. 2 and from the Cauchy-Schwarz inequality one can show generally that $-1 \leq a_{13} \leq 1$; if a turbulence model violates this, then it predicts "unrealizable" turbulence, meaning unphysical turbulence. For large $S_{13}$, the $k$–$\varepsilon$ model leads to unrealizable turbulence, while both the $k$–$\varepsilon$–$f_P$ model and WJ-EARSM (the 2D and 3D WJ-EARSMs are identical in 1D) are realizable even for large $S_{13}$. This is very desirable because large normalized velocity gradients are typically encountered in regions of non-equilibrium turbulence, e.g., in the vicinity of the rotor and in the wake shear layers. Indeed Réthoré (2009) noted large regions of unrealizable turbulence near the wind turbine, when using the standard $k$–$\varepsilon$ model. Also note that the $k$–$\varepsilon$–$f_P$ model (using $C_\mu = 0.03$) predicts a similar $a_{13}$ as the WJ-EARSM solution for large strains, while the main difference between the models is found for small strains.





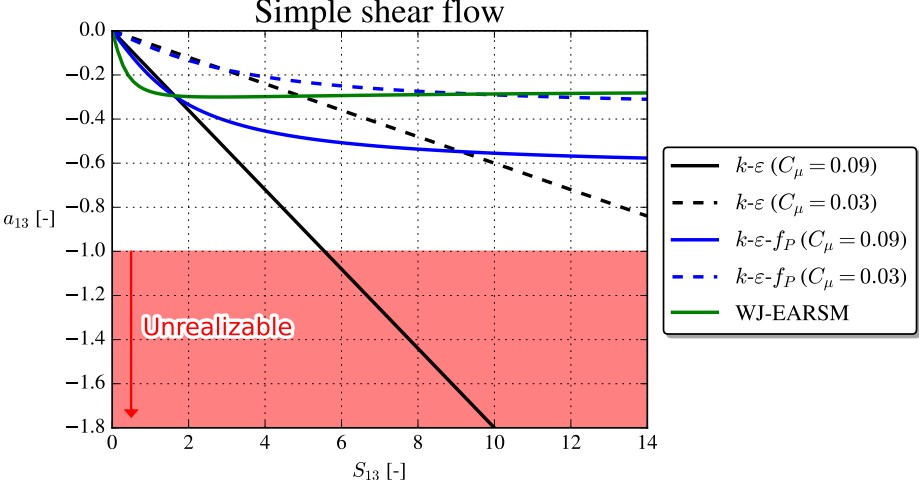

**Figure 2.** Analytical off-diagonal anisotropy in simple shear flow.

### 3.1 Homogeneous shear flow

Homogeneous shear flow (see review by Pope (2000), p.154-157) has a simple setup, but can be challenging to simulate and
is conceptually a strange case. The momentum equations are *not* solved and instead a constant velocity gradient is artificially
fixed at all times, see Figure 3. This is, indeed, rather unphysical since the velocity gradient should gradually decrease as
turbulence is created until the velocity gradient eventually becomes zero and turbulence dies out. Nevertheless, homogeneous
shear flow constitutes an interesting test case, because it only involves the solution of the turbulence transport equations and
turbulence closure, hence it is a "pure" test of the turbulence model. Moreover, free shear layers can locally be approximated
with homogeneous shear. Only EllipSys1D is used for this case.





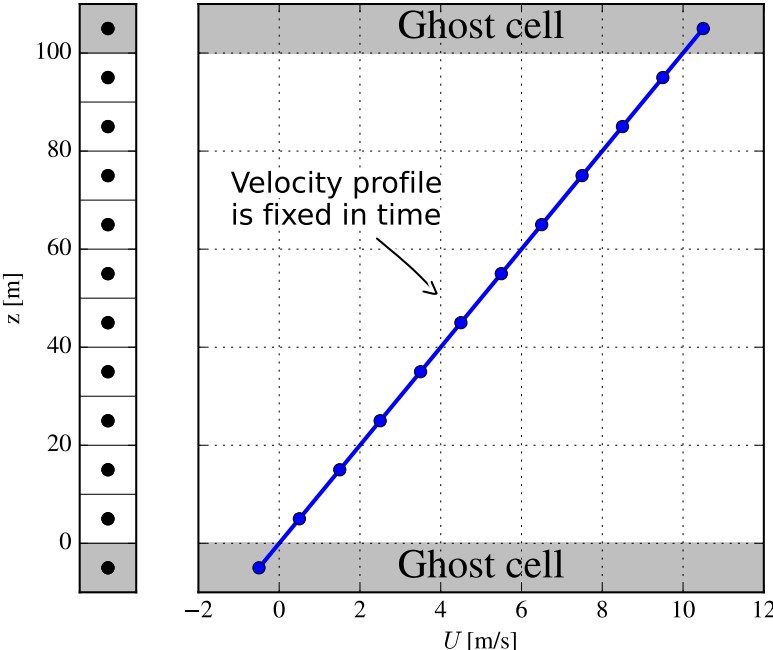

**Figure 3.** Mesh and the prescribed velocity profile in EllipSys1D for the homogeneous shear flow case.

In principle, homogeneous shear flow is an unsteady 0D case; the turbulence evolves identically at all positions in space, hence there is actually no need for a spatial discretization as shown in Fig. 3 (one could simply integrate the $k$ and $\varepsilon$ equations forward in time, given an initial turbulent state), but this extra complexity is chosen for the present case, since the goal is to verify the turbulence model implemented in a finite-volume CFD code. In practice this means that BCs need to be set at the
top and bottom of the domain: a fixed velocity, $U$, is used as illustrated in Fig. 3, while symmetry BCs are used for $k$ and $\varepsilon$. The implicit Euler scheme is used for time integration, while the second-order central scheme is used for diffusive terms.

The simulation parameters used are:

- Height of domain, $L_z = 100$ m. Uniformly spacing and 10 finite-volume cells used (if ghost cells are counted in, there are 12 cells), hence grid spacing is $\Delta z = 10$ m.

- Fixed velocity profile, $U(z) = Sz$, where $S = 0.1\ \mathrm{s}^{-1}$. This gives $U(z=0) = 0$ and $U(L_z) = 10$ m/s.

- Normalized timestep, $\Delta t^* \equiv \Delta t S = 0.1$ (in physical time $\Delta t = 1$ s). Ten subiteratons per timestep are used and the total normalized simulation time is $t^* = 80$ (in physical time $t = 800$ s).

- Initial turbulent state, $S\frac{k}{\varepsilon} = 3.4$ at $t^* = 0$. This non-dimensional quantity determines the evolution of the non-dimensional metrics, i.e., anisotropy components, production-to-dissipation ratio, etc. It is same initial turbulent state as used in the
homogeneous shear flow simulations by Bardina et al. (1983), Gatski and Speziale (1993), Girimaji (1996) and Wallin and Johansson (2002).





- The RANS model equations are here independent of $\rho$ and $\nu$ because no momentum equation is solved and the diffusion terms in the $k$ and $\varepsilon$ equations ($\mathcal{D}^{(k)}$ and $\mathcal{D}^{(\varepsilon)}$ in Eq. (2-3)) will be zero.

A nice feature of homogeneous shear flow for verification purposes is that it evolves to an asymptotic state and that an analytic solution exists for this state, e.g., the asymptotic production-to-dissipation ratio can be derived to be (see for example, Gatski and Speziale, 1993):

$$\frac{\mathcal{P}}{\varepsilon} = \frac{1 - C_{\varepsilon,2}}{1 - C_{\varepsilon,1}} \quad \text{(asymptotic limit } t \to \infty). \tag{21}$$

In 1D parallel flows, $\mathcal{P}/\varepsilon = -2a_{13}S_{13}$ and this gives a second equation for $\mathcal{P}/\varepsilon$. By inserting the expression for $a_{13}$ (the expression differs depending on the turbulence model), one can isolate and obtain a value for $S_{13} = \frac{1}{2}\frac{k}{\varepsilon}S$. This value can then
be used to obtain $S_{ij}$ and $\Omega_{ij}$ and hence the anistropy tensor. The asymptotic formulas are summarized in Table 3, where one can note that the asymptotic WJ-EARSM formulas only depend on $C_{\varepsilon1}$ and $C_{\varepsilon2}$, while the $k$–$\varepsilon$ formulas additionally depend on $C_\mu$. Figure 4 shows the time evolution of the WJ-EARSM as well as the theoretical asymptotic values for three common sets of coefficients ($C_{\varepsilon1}, C_{\varepsilon2}$). All simulations indeed go to the analytical asymptote, which gives some confidence in the implementation of the WJ-EARSM. For validation purposes we also include some LES (Bardina et al., 1983) and
experimental (Tavoularis and Corrsin, 1981) data in Fig. 4, which show that the Zeli constants perform better for all quantities except $a_{11}$. The choice of $c_2 = 5/9$ makes $a_{22} = 0$ for all simulations.

|  | $k$–$\varepsilon$ | WJ-EARSM |
| --- | --- | --- |
| $\frac{\mathcal{P}}{\varepsilon}$ | Eq. (21) | Eq. (21) |
| $S_{13}$ | $\frac{1}{2}C_\mu^{-1/2}\sqrt{\frac{\mathcal{P}}{\varepsilon}}$ | $N\left(\frac{12}{5}\frac{N}{\frac{\mathcal{P}}{\varepsilon}} - 4\right)^{-1/2}$ |
| $S\frac{k}{\varepsilon}$ | $2S_{13}$ | $2S_{13}$ |
| $a_{13}$ | $-2C_\mu S_{13}$ | $\beta_1 T_{13}^{(1)}$ |
| $a_{11}$ | 0 | $\beta_1 T_{33}^{(4)}$ |
| $a_{22}$ | 0 | 0 |
| $a_{33}$ | 0 | $-a_{11}$ |

**Table 3.** Analytic, asymptotic homogeneous shear flow formulas. In homogeneous shear flow, $S_{13} = S_{31} = \Omega_{13} = -\Omega_{31}$, while all other components are 0. When $\frac{\mathcal{P}}{\varepsilon}$ is known, one can use the direct definition of $N \equiv c_1' + \frac{9}{4}\frac{\mathcal{P}}{\varepsilon}$ instead of Eq. (12), see Wallin and Johansson (2000).



**Figure 4.** Simulation of homogeneous shear flow with WJ-EARSM (full line) and the model's analytical asymptotic values (dashed lines) with different sets of constants. The simulation data is extracted at $z/L_z = 0.45$, but the evolution is identical at all grid points.

## 3.2 Half-channel flow

The second reference flow is the fully developed, steady-state half-channel flow, aka. pressure driven boundary layer (PDBL) flow, and can be solved in 1D using a grid as sketched in Fig 5. The lower BC is a rough wall implemented as in Sørensen et al.

(2007), while the upper BC uses symmetry, hence turbulence now depends on z-coordinate contrary to in the homogeneous shear flow case. The flow is driven by a constant streamwise pressure gradient force, which is an input parameter for the



simulation (if one does not use such a forcing, then the rough wall will extract momentum from the flow and the velocity eventually becomes zero throughout the domain).

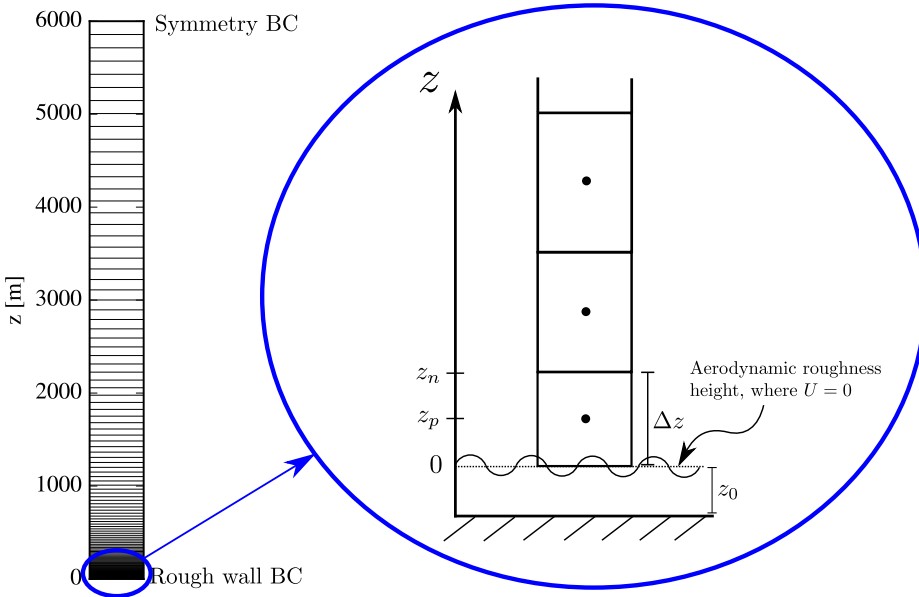

**Figure 5.** Half-channel EllipSys1D setup. The sketch to the right shows convention of coordinate system used for rough wall simulations.

The input parameters for the simulations are:

• The domain height, $L_z = 6000$ m.

• Aerodynamic roughness height, $z_0 = 0.03$ m.

• First cell height, $\Delta z = 0.10$ m.

• 192 cells (+ 2 ghost cells) and the grid is stretched using the hyperbolic tangent function (Thompson et al., 1985).

• The streamwise pressure gradient force per unit mass, $F_p = 1.5 \cdot 10^{-5}$ m/s$^2$. One can show that the squared friction

velocity then becomes, $u_*^2 \equiv -\overline{u'w'}_s = F_p L_z = 0.09$ m/s.

• The flow is independent of $\nu$ because a rough wall BC is used and the flow is fully turbulent, hence $\nu \ll \nu_t$.

The region in the lower part of the domain is known as the "log-layer", which is characterized by equilibrium turbulence, i.e., $\frac{\mathcal{P}}{\varepsilon} \approx 1$. As in the homogeneous shear flow case, we can use this ratio to obtain analytical results for shear and anisotropy, see Table 4.





| | $k-\varepsilon$ | WJ-EARSM |
|---|---|---|
| $\frac{\mathcal{P}}{\varepsilon}$ | 1 | 1 |
| $S_{13}$ | $\frac{1}{2}C_\mu^{-1/2}\sqrt{\frac{\mathcal{P}}{\varepsilon}}$ | $N\left(\frac{12}{5}\frac{N}{\frac{\mathcal{P}}{\varepsilon}}-4\right)^{-1/2}$ |
| $a_{13}$ | $-2C_\mu S_{13}$ | $\beta_1 T_{13}^{(1)}$ |
| $a_{11}$ | 0 | $\beta_1 T_{33}^{(4)}$ |
| $a_{22}$ | 0 | 0 |
| $a_{33}$ | 0 | $-a_{11}$ |

**Table 4.** Analytical values in the log-layer of the half-channel flow.

The flow profiles with the analytical log-layer values are plotted in Fig 6. Both sets of model constants, Table 1 and 2, are used for the $k-\varepsilon$ model, which explains the different $U$-profiles. A spike in the $k$-profiles is seen near the wall, which is a well known problem, see Blocken et al. (2007), but the value of $k(z \approx 0)$ is nevertheless close to the equilibrium value $k_{\mathrm{eq}} = \frac{u_*^2}{C_\mu^{1/2}}$ for both models (again reminding that $C_\mu$ differs between the two sets of model constants, compare Table 1 and 2). The kinematic wall shear stress, $\overline{u'w'}(z=0)$, is close to $-u_*^2$ for both models confirming that the pressure gradient force is applied

correctly and the respective analytical log-layer solutions are also approximately obtained for both models at $z/L_z < 0.3$. One can notice that a feature of the WJ-EARSM is that the normal streamwise and vertical anisotropies are non-zero; this means $\overline{u'u'} > \overline{v'v'} = \frac{2}{3}k > \overline{w'w'}$. This behavior is also seen in the neutral ASL (Panofsky and Dutton, 1984), which we return to in Sect. 4.

**Figure 6.** Half-channel simulation results (full lines) and analytical log-layer solutions (dashed lines).

## 3.3 Square duct flow

The square duct geometry and boundary conditions are shown in Fig. 7. Similar to the half-channel flow, the flow is driven by a streamwise pressure gradient, but the difference is that the duct flow has walls on all four sides. Due to symmetry, we only simulate the lower left quadrant of the duct. As this paper focuses on high-Re turbulence models, we choose to model the walls as rough walls instead of smooth walls, which are traditionally used in square duct flow simulations.



Although, the fully developed square duct flow might appear as a 2D problem, it in fact features a full 3D flow field, due to
the secondary corner flows, also sketched in Fig. 7, which were first observed experimentally by Nikuradse (1930) and later
with DNS by Gavrilakis (1992). The secondary motions are only on the order of $\frac{V}{U} \sim \frac{W}{U} \sim \mathcal{O}(10^{-2})$ but still have a notable
effect on the bulk flow as they transport momentum from the center of the duct toward the corners. Perhaps the most interesting
aspect of square duct flow from a turbulence modeling perspective is that linear EVMs are unable to predict the secondary
corner flows, because the secondary motions are caused by the normal anisotropy components, which are zero in linear EVMs
for fully developed flow, see Eq. (5) and discussions by Menter et al. (2009); Emory et al. (2013). However more sophisticated
turbulence models such as EARSM and uncertainty quantification models (Emory et al., 2013) *are* able to predict this physical
phenomenon.

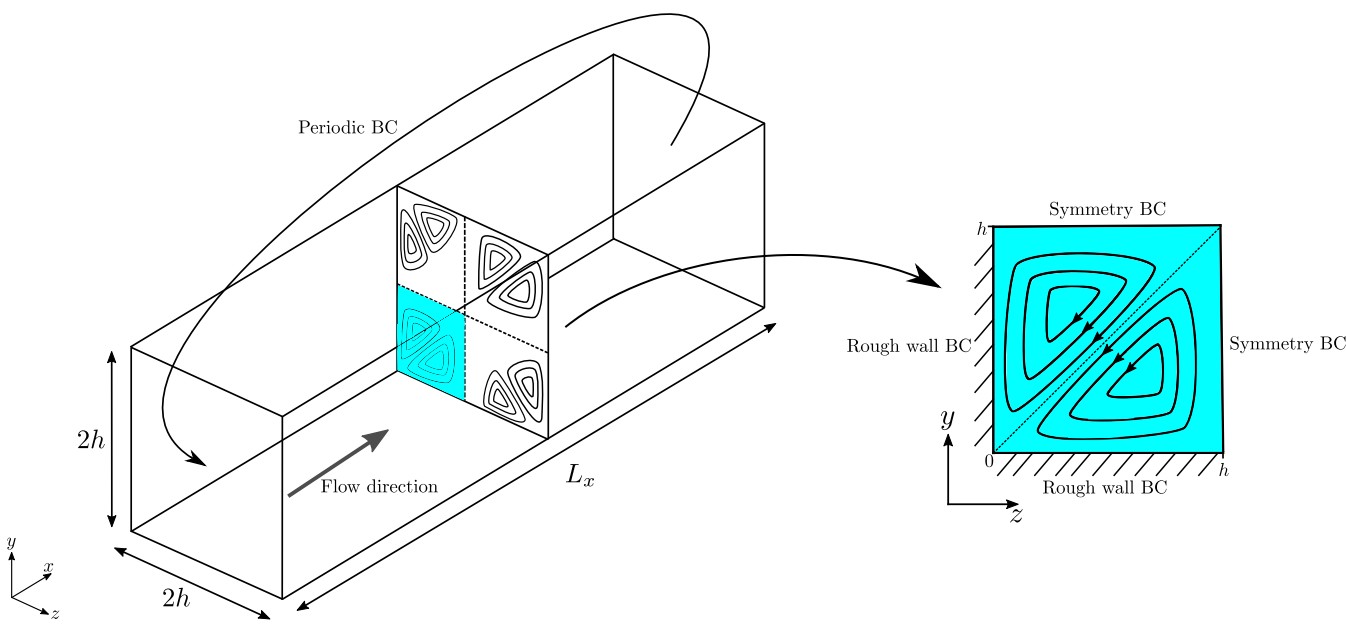

**Figure 7.** Square duct geometry and a cross-section showing the secondary corner flows. Sketch made with inspiration from Wu et al. (2016).

In this paper, we simulate a fully turbulent square duct flow (high-Re) with rough wall BCs, hence the flow is independent
of $\nu$ similar to the half-channel flow. Currently, the DNS of Pirozzoli et al. (2018) is the most turbulent DNS available ($\mathrm{Re}_\tau \equiv$
$\frac{\langle u_* \rangle h}{\nu} = 1055$), so this is chosen for reference, although it uses smooth walls and might not correspond exactly to our high-Re
case. For this reason the RANS and DNS should not be compared directly, but the DNS will at least show typical characteristics
and the order of magnitude to be expected. The parameters for the RANS simulation are given in Table 5 and a rectilinear
32x64x64 grid is used, which is stretched towards the walls to obtain a first wall-adjacent cell height on the order of the
roughness length (no stretching used in streamwise direction).





| $h$ [m] | $L_x/h$ [-] | $z_0/h$ [-] | $\langle u_* \rangle$ [m s$^{-1}$] |
|---------|-------------|-------------|-----------------------------------|
| 640 | 4 | $7.81 \cdot 10^{-5}$ | $1.24 \cdot 10^{-2}$ |

**Table 5.** Parameters used for simulation of square duct flow in EllipSys3D. The brackets $\langle \rangle$ signifiy the average over the wall.

The streamwise and vertical velocity ($U$ and $V$ components) are shown in Fig. 8, which show that the 2D WJ-EARSM is indeed capable of predicting secondary flows similar to the DNS of Pirozzoli et al. (2018). Both Fig. 8 and 9 clearly shows that the prediction of the secondary flow is necessary to capture the correct shape of the $U$-distribution. In contrast, the standard $k$–$\varepsilon$ model predicts zero vertical velocity and therefore no secondary flow.

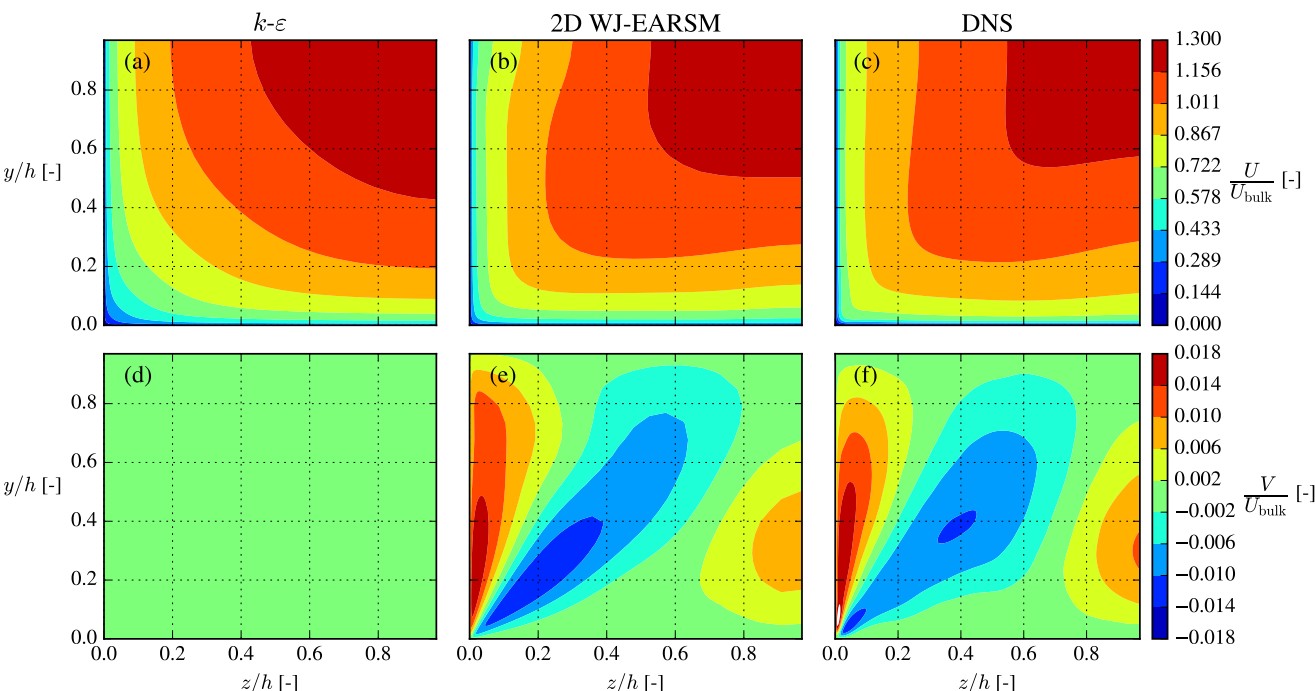

**Figure 8.** Streamwise (first row) and vertical (second row) velocity contours in the lower left quadrant of the square duct. Normalized by the bulk velocity, $U_{\text{bulk}} = \frac{1}{A} \int U dA$.

The 3D WJ-EARSM performed similar as the 2D WJ-EARSM, although with a slightly weaker secondary flow, which can
be seen in Fig. 9, where the velocity profiles are extracted on the corner bisector line (the diagonal line). Both WJ-EARSMs predict similar profiles to the DNS, although without the near corner peak, which according to Pirozzoli et al. (2018) is caused by scale separation (this phenomena only occurs at higher $\text{Re}_\tau$ and is thus not visible in the earlier DNSs by Gavrilakis (1992) and Huser and Biringen (1993)) and this effect is not captured by RANS.





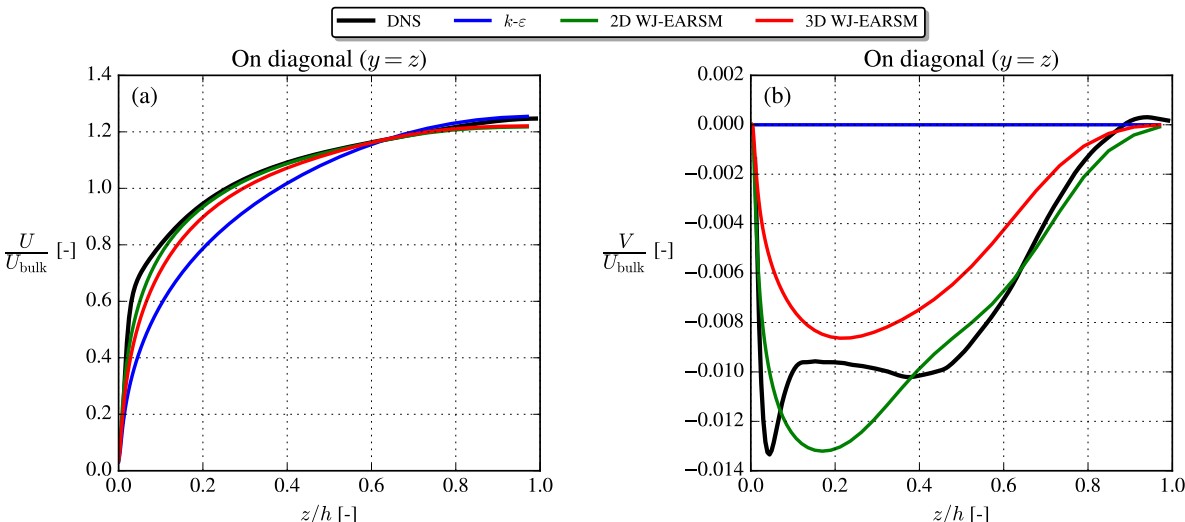

**Figure 9.** Streamwise velocity (a) and vertical velocity (b) on the corner bisector line. Results extracted at $x/L_x = 0.5$.

This concludes the verification studies, where the WJ-EARSM has been seen to give expected results for three canonical
flows. Furthermore, the last case of square duct flow clearly demonstrates that EARS models are able to predict physical
phenomena that two-equation models based on the linear Boussinesq hypothesis cannot do.

## 4   Single wind turbine wake

This section concerns the application of the EARS model to a single wind turbine wake. The numerical CFD setup is similar
to that used in many previous RANS studies, e.g., van der Laan (2014); van der Laan et al. (2021); Baungaard et al. (2022):
the RANS equations are solved using the SIMPLE method using a modified Rhie–Chow algorithm (Troldborg et al., 2015),
while the convective terms are discretized with the QUICK scheme, see more details in Sørensen (1995). To model the wind
turbine, an AD with uniform distribution of forces is used (the thrust forces are fixed and no tangential forces are present),
and the forces are transferred to the rectilinear flow domain with the intersectional method of Réthoré et al. (2014). The flow
domain has a finely resolved "wake domain" in the center with uniform spacing of $D/8$, and the grid is stretched outwards in
all directions from this using the hyperbolic tangent method of Thompson et al. (1985), see Fig. 10.

The case simulated is similar to the case used by Hornshøj-Møller et al. (2021), namely a single V80 turbine subject to
neutral inflow, see Table 6. The authors of the aforementioned study have provided LES data to us, which will be used as a
reference in the following. It should be noted that their RANS simulations use ASL inflow (like we shall also use for our RANS
simulations), while their LES is based on PDBL inflow. Although there will be differences between ASL and PDBL inflow,
the latter is likely a good approximation of the former in the lower part of the domain, but its bias on wake simulations could
be a subject for future studies.



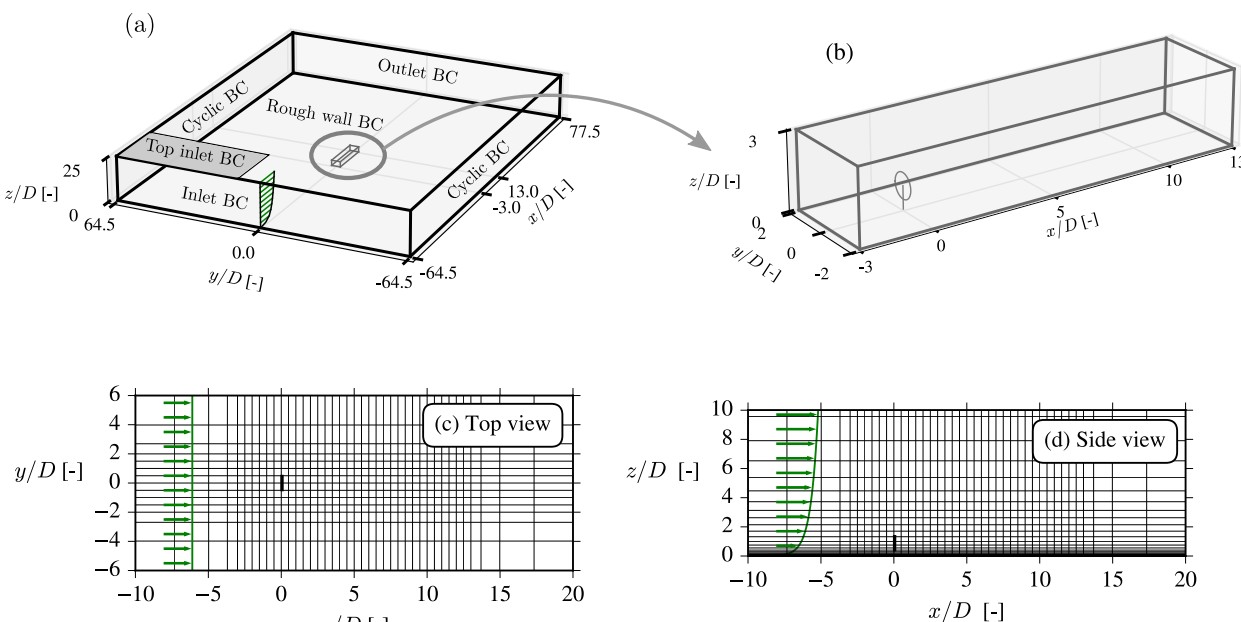

**Figure 10.** Flow domain (a), finely resolved wake domain (b), xy-cut at hub height (c) and xz-cut at centerline (d). Every fourth cell is displayed in (c,d) and green arrows show velocity profile (not to scale).

| Inflow | | | | Turbine | | | | | |
|---|---|---|---|---|---|---|---|---|---|
| $U_{\mathrm{ref}}$ [m/s] | $I_{\mathrm{ref}}$ [%] | Atmospheric model | $D$ [m] | $z_{\mathrm{ref}}$ [m] | Force distribution | Tangential forces | $C_T$ [-] | Control |
| 8.0 | 5.7 | Neutral ASL | 80 | 70 | Uniform | No | 0.77 | $C_T$ fixed |

**Table 6.** Simulation parameters for the single wake V80 case. Note, that the LES uses neutral PDBL for the atmospheric model.

### 4.1 Inflow

The neutral ASL inflow profile (Panofsky and Dutton, 1984) is prescribed at the inlet BC and top of the domain:

$$U(z) = \frac{u_*}{\kappa}\ln(z/z_0) \quad , \quad k(z) = \frac{u_*^2}{\sqrt{C_\mu}} \quad , \quad \varepsilon(z) = \frac{u_*^3}{\kappa z}. \tag{22}$$

This type of inflow is routinely used in wind energy applications, and for a wake simulation it can be adopted to give a desired hub height velocity, $U_{\mathrm{ref}}$, and hub height turbulence intensity (TI), $I_{\mathrm{ref}}$, by adjusting $u_*$ and $z_0$ (van der Laan et al., 2015b). One could alternatively adjust $C_\mu$ instead of $z_0$ to obtain a desired $I_{\mathrm{ref}}$, but this was shown by van der Laan (2014) to give



inconsistent results with the $k$–$\varepsilon$-$f_P$ model and NLEVMs, so this practice is not recommended. Although the adjusted $u_*$ and $z_0$ do not correspond to the physical values at the site, it is of higher priority to have the correct $U_{\mathrm{ref}}$ (and thereby correct thrust coefficient) and $I_{\mathrm{ref}}$.

In the freestream, the neutral ASL profiles, Eq. (22), should satisfy $Dk/Dt = 0$ and $D\varepsilon/Dt = 0$ to be in balance and mitigate development of the inflow profiles in the streamwise direction. There will inevitably be a slight development, see Blocken et al. (2007), and for this reason a long domain is used to ensure fully developed profiles at the entrance of the wake domain. To satisfy the balance criteria, the turbulence constants should follow the relation (Richards and Hoxey, 1993):

$$C_{\varepsilon,1} = C_{\varepsilon,2} - \frac{\kappa^2}{\sqrt{C_\mu}\sigma_\varepsilon}. \tag{23}$$

Indeed, both sets of constants in Table 1 and 2 satisfy Eq. (23).

In the neutral ASL, we have $\mathcal{P}/\varepsilon = 1$, similar to the log-layer of the half-channel flow but at all heights. By utilising the WJ-EARSM the equilibrium relations between the $c_1$ constant and various other variables, as shown in Fig. 11, can be derived analytically. For example, the standard value of $c_1 = 1.8$, which is used in this paper, gives $C_\mu = 0.087$ (the "equilibrium value" mentioned in Sect. 2.3.1). A closer agreement of the velocity standard deviation ratios, $\sigma_v/\sigma_u$ and $\sigma_w/\sigma_u$, is seen with the WJ-EARSM, when benchmarking against the ASL ratios of Panofsky and Dutton (1984); the $k$–$\varepsilon$ and $k$–$\varepsilon$–$f_P$ models have $\sigma_v/\sigma_u = \sigma_w/\sigma_u = 1$, while the WJ-EARSM has $\sigma_v/\sigma_u = 0.85$ and $\sigma_w/\sigma_u = 0.68$ for $c_1 = 1.8$. In fact, the WJ-EARSM is underpredicting the streamwise fluctuations due to the simplification introduced by setting $c_2 = 5/9$ (this is also the case for half-channel flow, see Wallin and Johansson (2000)), resulting in the observed overprediction of the $\sigma_v/\sigma_u$ and $\sigma_w/\sigma_u$ ratios. Other $c_1$ values (three other choices than the standard value are marked with dots in Fig. 11) will enhance/decrease the ASL anisotropy, see details in Appendix B.





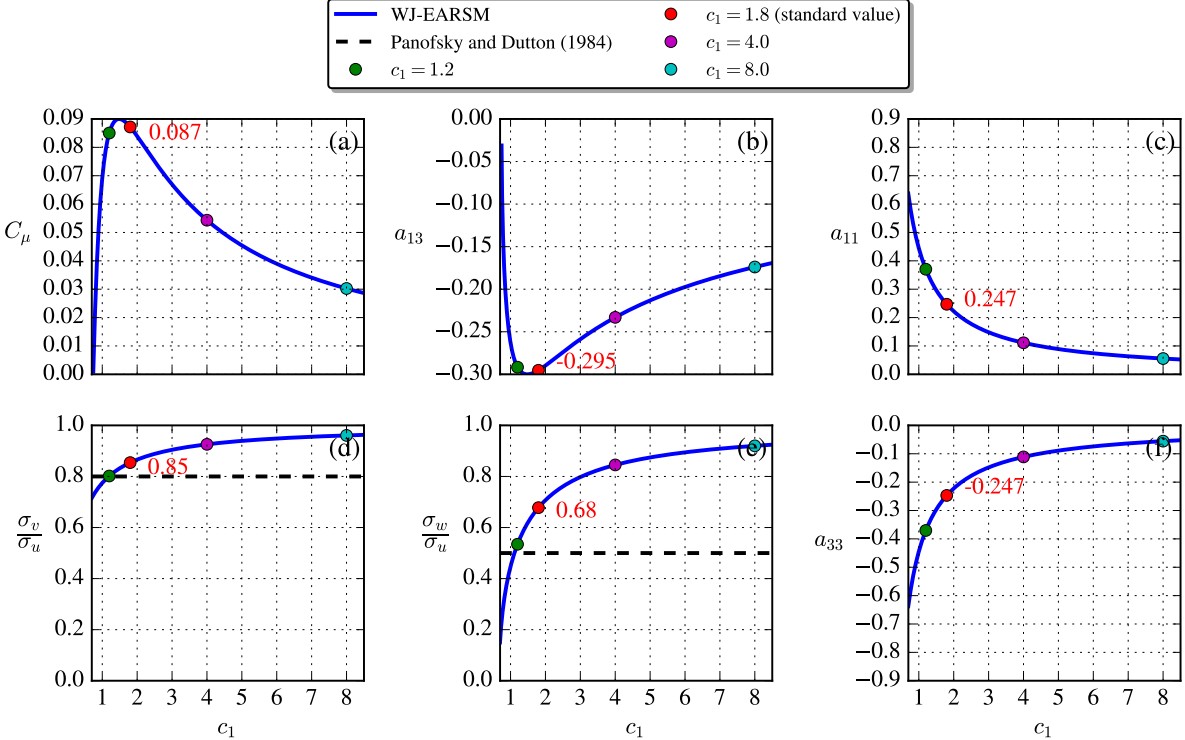

**Figure 11.** WJ-EARSM dependence on $c_1$ in the neutral ASL ($\mathcal{P}/\varepsilon = 1$).

The inflow profiles of the LES and RANS simulations are shown in Fig. 12. Since the $k$–$\varepsilon$ and $k$–$\varepsilon$–$f_P$ models are identical in the freestream, which is also true for the 2D and 3D WJ-EARSMs, then only one of the other is shown in the figure. The LES is a driven with a streamwise pressure gradient and thus differs in the stress profiles compared to the pure ASL profiles used in our RANS, but $U_{\mathrm{ref}}$ and $I_{\mathrm{ref}}$ do match. As in the half-channel case, the velocity shear differs due to the different $\kappa$ and $C_\mu$ used in the $k$–$\varepsilon$–$f_P$ model and WJ-EARSM, and the stress profiles are also different between the two because of the anisotropic nature of WJ-EARSM. In terms of turbulence anisotropy, quantified as the distribution of TKE between the normal stress components in Fig. 12, the WJ-EARSM is clearly closer to the LES data as was also expected from the analytical results in Fig. 11.





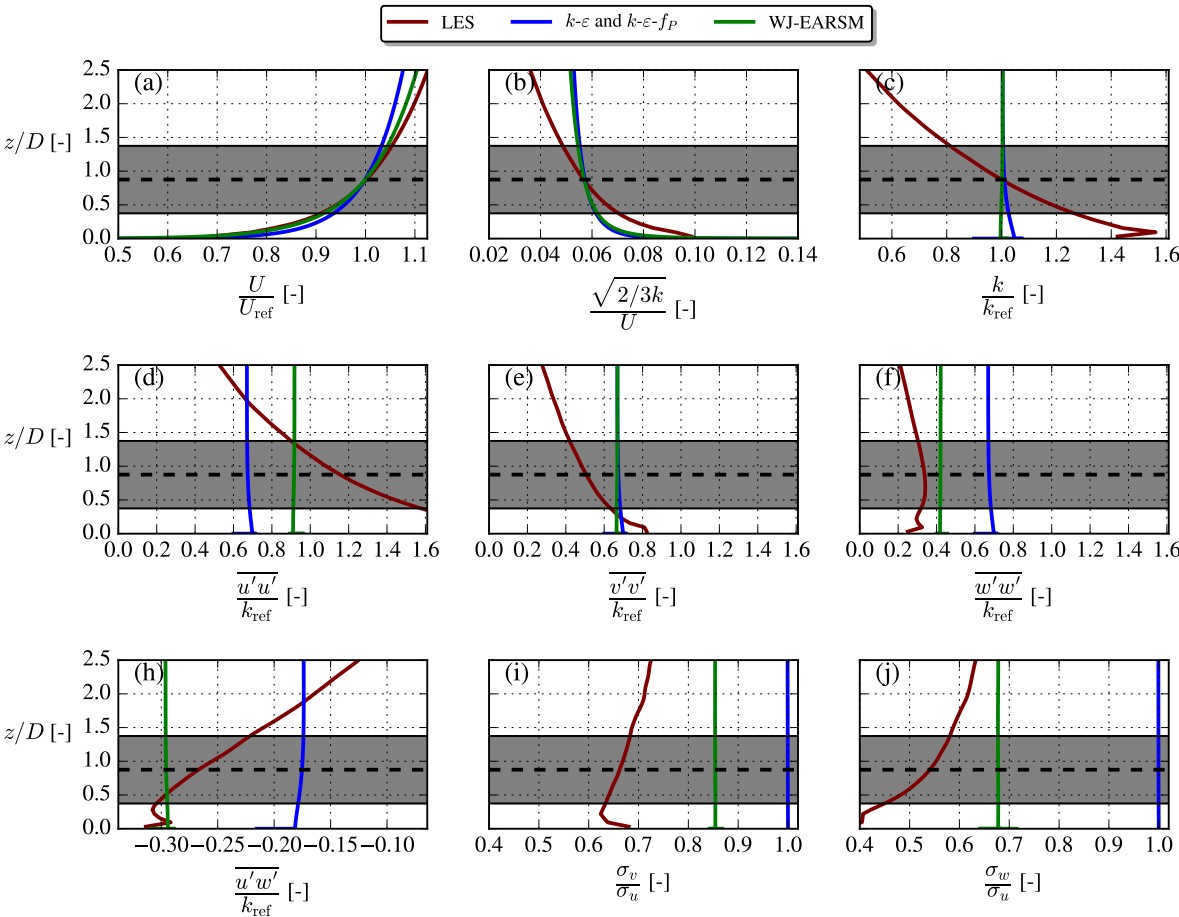

**Figure 12.** Inflow profiles for the single wake V80 case.

Eigendecomposition of the Reynolds stress tensors can be used to describe the turbulence state through its three real eigenvalues, which describe the fluctuations in the three orthogonal, principal directions. Several techniques (e.g., eigenvalue map, invariant map, Barycentric map and Lumley triangle) combine the eigenvalues and visualize them with 2D maps; in Fig. 13 the Barycentric map is used with the RGB color scheme of Emory and Iaccarino (2014). Both RANS models use ASL inflow, hence the turbulence state is the same at all heights, whereas the LES state varies with height due to its PDBL inflow. The

$k$–$\varepsilon$–$f_P$ turbulence is mostly isotropic (the $x_{3c}$ corner) as is expected for all Boussinesq-type models because normal stresses are $\overline{u'_\alpha u'_\alpha} = \frac{2}{3}k$, whereas the WJ-EARSM turbulence is perturbed more towards 2D turbulence (the line connecting $x_{2c}$ and $x_{1c}$) and thereby closer to LES turbulence.





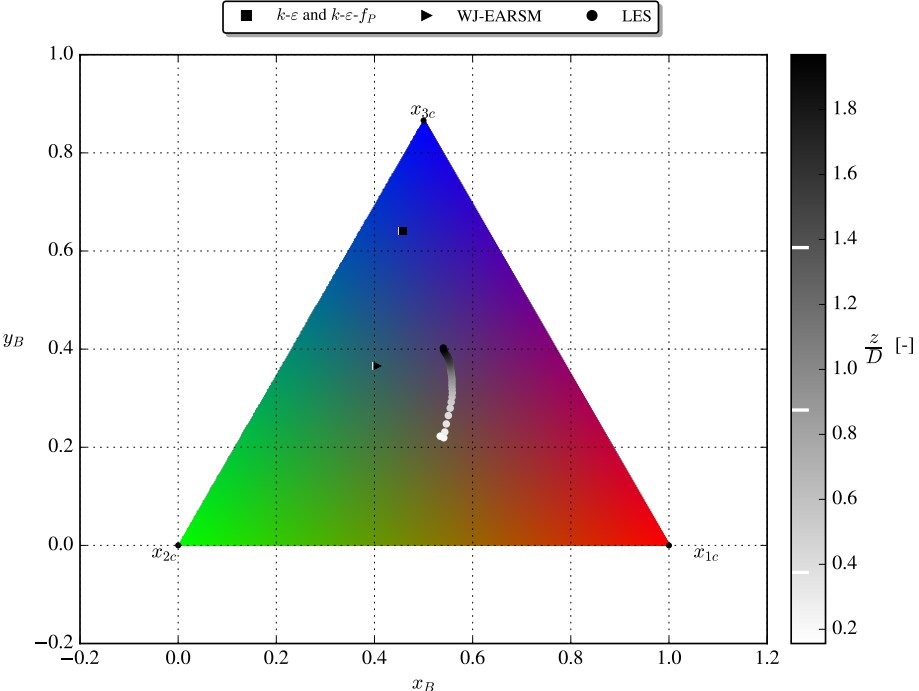

**Figure 13.** RGB-colored Barycentric triangle and RANS/LES inflow data colored by height (white lines mark $z_{ref} - R$, $z_{ref}$ and $z_{ref} + R$, respectively).

## 4.2 Velocity and turbulence intensity

Wake data in form of velocity and TI contours at hub height and profiles at three downstream positions are shown in Fig. 14-
15. In addition to the 2D WJ-EARSM, results using the 3D WJ-EARSM with/without tangential AD forces are also shown in Fig.15, and the first conclusion to draw from this is that the 2D and 3D versions of the WJ-EARSM give similar wake profiles and we will therefore focus on the simpler 2D WJ-EARSM in the following. A simple diffusion correction to the WJ-EARSM was suggested by Wallin and Johansson (2000) to correct the model in regions with low normalized velocity gradients (e.g., at the top of the half-channel), but it only has a small effect on velocity deficit and TI as seen in Fig. 15, so it will not be
considered in the following discussions. Finally, results of the standard $k$–$\varepsilon$ model are also shown in Fig. 15 to show its overly diffusive behaviour.

Overall the wake velocity contours in Fig. 14 appear similar, while the TI contours of the WJ-EARSM are improved over the $k$–$\varepsilon$–$f_P$ model; however similar to the $k$–$\varepsilon$–$f_P$ model it still fails to predict the TI delay in the near wake seen in the LES, which is also clearly visible in the disk-averaged TI recovery profiles in Fig. 16b. Also, there is a TI induction zone in both
RANS simulations, which is not present in the LES; these two effects seem to be a general issue of $k$–$\varepsilon$ based RANS models as this was also observed with the standard $k$–$\varepsilon$, realizable $k$–$\varepsilon$ and RNG $k$–$\varepsilon$ models by Hornshøj-Møller et al. (2021).



Considering the wake profiles in Fig. 15, it is clear that WJ-EARSM produces more "top-hat shaped" profiles similar to the classic Jensen model (Jensen, 1983). This has also been seen with other EARSMs (van der Laan, 2014) and with full differential RSMs (Cabezón et al., 2011; Tian et al., 2019). One can either apply the previously mentioned diffusion correction

or increase the $c_1$ constant to obtain a more Gaussian shaped profile for the WJ-EARSM, but the latter is not recommended as it will deteriorate the ASL anisotropy, see Appendix B. Another view is that the top-hat shaped profile is a consequence of not taken various physical phenomena into account, e.g., large scale atmospheric turbulence and wake instabilities, and by applying a unidirectional wind direction applied in our idealized RANS setup (in the transient LES there will be a varying instantaneous wind direction throughout the simulation). These type of effects could be interpreted as a Gaussian filter on the wake profiles,

see discussion in Appendix C. Note that the $k$–$\varepsilon$–$f_P$ model already accounts for at least the wind direction effect as it has been calibrated with LES that includes a wind direction distribution (van der Laan, 2014) and that it could be recalibrated to obtain a better match with the velocity deficit of the present LES case (since the latter LES was simulated with a different solver and AD implementation).

Lastly, we want to emphasize that the turbulence model is not the only responsible for the wake results: the same turbulence

model applied for the same case but with different codes/solvers can yield significantly different results, as can be seen in the comparison of the $k$–$\varepsilon$–$f_P$ results in Fig. 16. This reminds us to be careful with general conclusions on which is the better turbulence model.

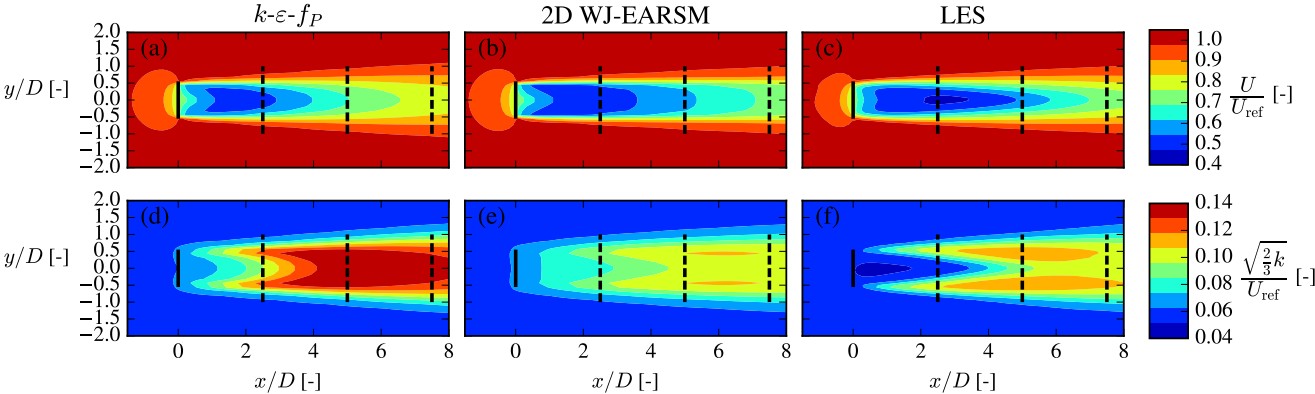

**Figure 14.** Streamwise velocity (upper row) and TI (lower row) contours at hub height for the single wake V80 case. Full lines mark ADs, while dashed lines mark where wake profiles are extracted.



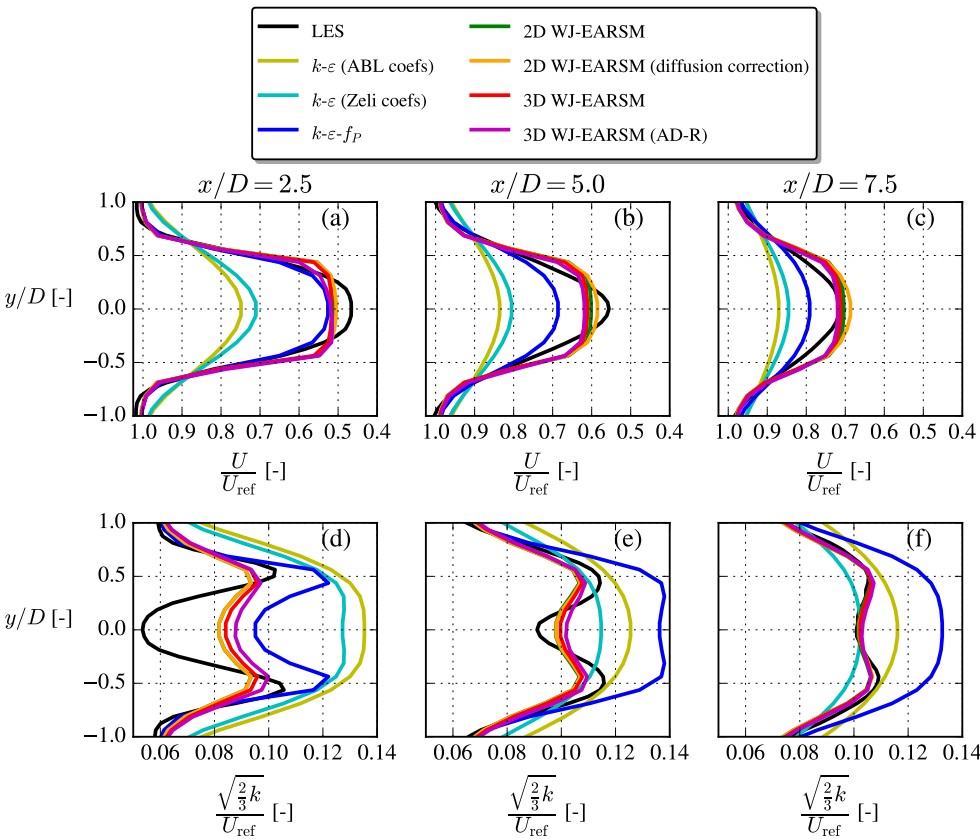

**Figure 15.** Streamwise velocity (upper row) and TI (lower row) profiles extracted at various donwstream positions for the single wake V80 case.



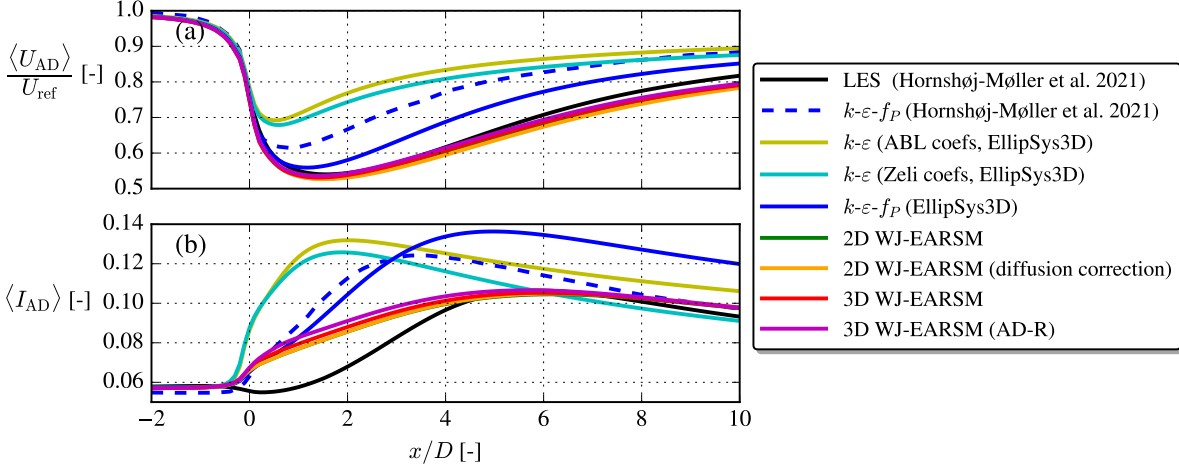

**Figure 16.** Disk-averaged streamwise velocity and TI for the single wake V80 case.

## 4.3 Stresses

For more insights on the wake mixing and turbulence, we now turn to the second order statistics of turbulence, namely the
individual Reynolds stress components. The normal components are shown in Fig. 17, which shows that the WJ-EARSM
correctly dampens the lateral and vertical components, which are else overestimated severely by the $k$–$\varepsilon$–$f_P$ model due to its
Boussinesq closure. This also explains the lower TI of the WJ-EARSM in Fig. 14-16 as the TI is composed by the sum of the
normal components. On the other hand, the normal stress contours seem too elongated in the streamwise direction with the
WJ-EARSM, for example the low $\overline{u'u'}$ core in the center of the wake in Fig. 17b extends too far downstream compared to the
LES data in Fig. 17c, which is possibly connected with the decreased turbulence mixing also causing the top-hat profiles of
wake deficit shown in Fig. 15. Increasing the $c_1$ constant will alleviate this specific issue, see Appendix B.



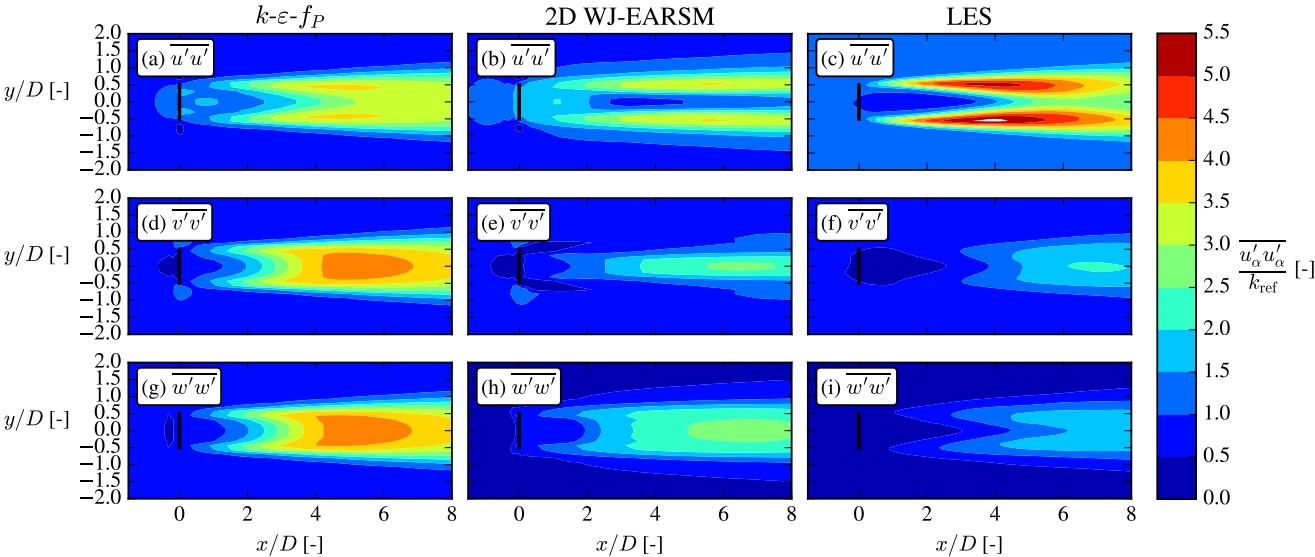

**Figure 17.** Normal stresses at hub height for the single wake V80 case.

Transport of $U$-momentum by turbulence, $-\partial \overline{u'u_j}/\partial x_j$, from the ambient high speed surroundings to the low speed wake region is mainly determined by the cross components of the Reynolds stress tensor, $\overline{u'v'}$ and $\overline{u'w'}$, i.e., the lateral and vertical turbulent fluxes of $U$, respectively, which are shown as contours in Fig. 18. Both are larger in absolute magnitude for the $k$–$\varepsilon$–$f_P$ model compared to WJ-EARSM, leading to larger gradients of the stresses and explaining the increased wake recovery of the former, but when comparing to the LES data, it can be noticed that the magnitude of $\overline{u'w'}$ is overestimated, hence the overestimated wake recovery of the $k$–$\varepsilon$–$f_P$ model in Fig. 16. On the other hand, the WJ-EARSM underestimates the magnitude of the fluxes, especially notable in the $\overline{u'v'}$ contours, hence the slight underestimation of wake recovery compared to LES in Fig. 16.

To conclude, we see some advantages but also disadvantages with using WJ-EARSM over the $k$–$\varepsilon$–$f_P$ model for prediction of the Reynolds stresses, which also have direct consequences for the prediction of the velocity deficit and turbulence intensity.



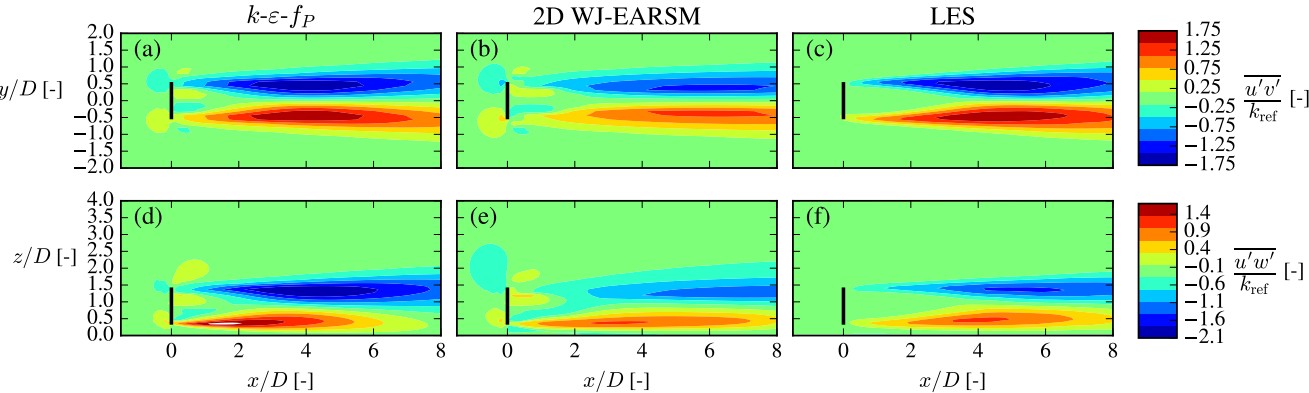

**Figure 18.** Lateral $U$-momentum flux at hub height xy-plane (upper row) and vertical $U$-momentum flux at the center xz-plane (lower row) for the single wake V80 case.

## 4.4 Turbulence state

In the inflow section, Sect. 4.1, it was shown that the turbulence state of the freestream turbulence was mainly isotropic in the $k$–$\varepsilon$–$f_P$ model, while both the WJ-EARSM and LES data were more perturbed towards 2D turbulence (the lower edge in the

Barycentric triangle). This is also seen in RGB-colored xy-plane at hub height in Fig. 19, where one can see that the ambient flow of the $k$–$\varepsilon$–$f_P$ model is predominantly colored blue and hence isotropic. As also alluded in Sect. 4.1 the ambient WJ-EARSM turbulence is more oblate (left edge of Barycentric triangle), while the LES is more prolate (right edge of Barycentric triangle), hence the green and purple coloring, respectively, of the ambient flow in Fig. 19. In the wake shear layers, both RANS models predict perturbations towards 2D turbulence, while LES is more perturbed towards 1D turbulence (the $x_{1c}$

corner). These observations fit with the increase of $\overline{u'u'}$ and strong damping of $\overline{v'v'}$ and $\overline{w'w'}$ in the LES shown in Fig. 17. To improve the prediction of the turbulence state, we therefore suspect that normal stress predictions are essential, especially the ratios of those - the WJ-EARSM definitely has some improvement from the overly isotropic $k$–$\varepsilon$–$f_P$ model but does not capture the completely right ratio between normal stresses. We note that a resolution of 20 cells per diameter was used in the reference LES, which might be sufficient for first order statistics and shear stresses, but could bias the normal stresses and

therefore also the turbulence state.

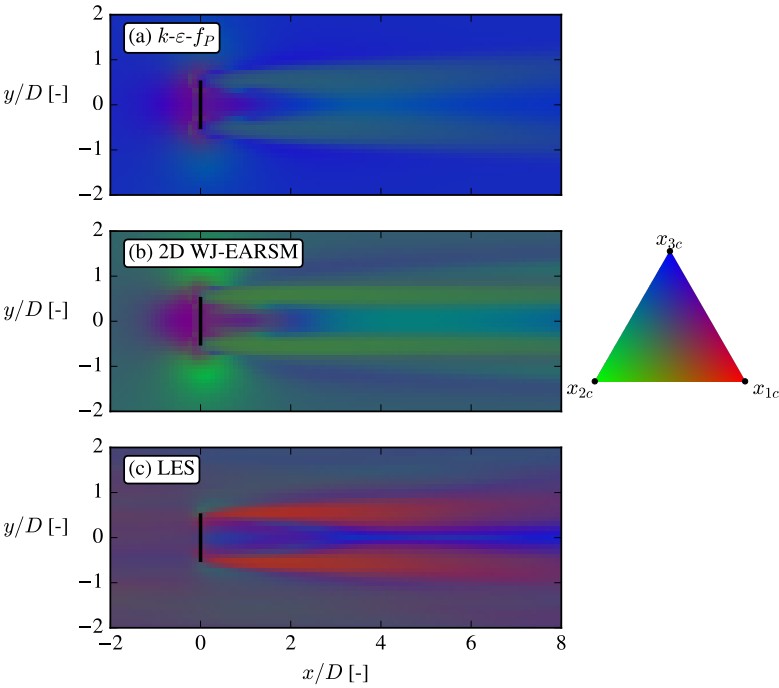

**Figure 19.** RGB-colored turbulence componentiality at hub height for the single wake V80 case.

## 5 Aligned row of wind turbines

To test the WJ-EARSM in a wind farm scenario, we simulate the lower row in the TotalControl rot90 reference wind farm, which consists of ten aligned wind turbines with $5D$ inter-spacing, see Fig. 20. The DTU-10MW turbines of the wind farm are modeled with the scaling AD method with 1D momentum control van der Laan et al. (2015a) and including tangential forces
(hence there will be wake swirl), and the thrust, rotational speed and power curves are taken from the DTU-10MW report (Bak et al., 2013). We compare the results with LES conducted at KU Leuven (the "PDk90 case", Sood and Meyers (2020)), where the turbines were modeled with an actuator surface (AS) model coupled to an aeroelastic code, hence the LES and RANS are not directly comparable but nevertheless gives a reference to compare against. Due to the natural streaks appearing in LES, we choose to calculate $U_{\mathrm{ref}}$ and $I_{\mathrm{ref}}$ with planar averages of the LES data in a region upstream of the lower row (see Fig. 20)
rather than using the time-averaged precursor profiles. The overview of the simulation parameters are given in Table 7. The numerical setup of the RANS simulation is identical to the one used for the single wake case, except that tangential forces and non-uniform thrust forces are applied on the AD, the domain is scaled with the new rotor diameter and the wake region is extended to encompass all ten turbines.





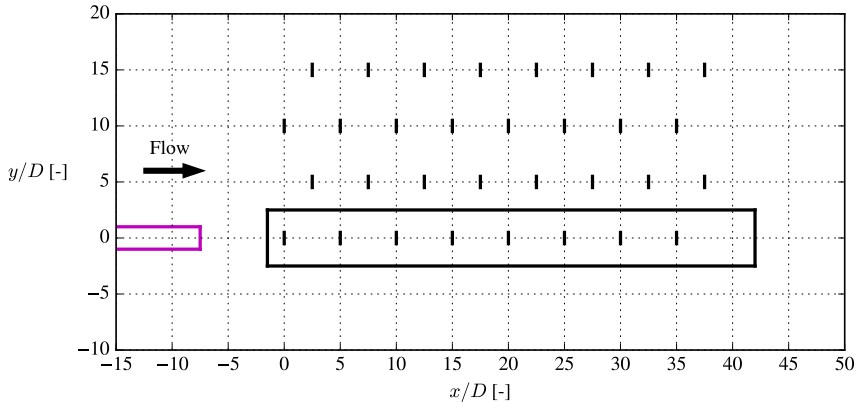

**Figure 20.** Layout of the TotalControl rot90 reference wind farm (Andersen and Troldborg, 2020). We simulate the flow in the lower row in this paper (black box) and calculate $U_{ref}$ and $I_{ref}$ with planar averages of the LES data in an area in front of the row (purple box).

| Inflow | | | Turbine | | | | |
|---|---|---|---|---|---|---|---|
| $U_{ref}$ [m/s] | $I_{ref}$ [%] | Atmospheric model | $D$ [m] | $z_{ref}$ [m] | Force distribution | Tangential forces | Control |
| 9.8 | 4.5 | Neutral ASL | 178.3 | 119 | Scaling | Yes | 1D momentum method |

**Table 7.** Simulation parameters for the aligned row case TotalControl case. The AD scaling and 1D mom'm control methods are described in detail by van der Laan et al. (2015a).

The velocity contours in Fig. 21 show that both the $k$–$\varepsilon$–$f_P$ model and WJ-EARSM qualitatively share some of the same flow features as the LES, e.g., a faster wake recovery is seen on the left side of the first wake (seen from upstream) because of the combined effect of wind shear and wake rotation (aka. swirl) and they predict the largest wake deficit in the second wake. Neither of these effects are predicted by the standard $k$–$\varepsilon$ model, but it can capture other phenomena as for example the induction zones and the expanding wake tube surrounding the whole row. Fig. 23a shows the streamwise velocity along the axial line of the ADs and shows that the WJ-EARSM is closer to the LES in the near wake of each turbine, while the $k$–$\varepsilon$ and $k$–$\varepsilon$–$f_P$ models are better at the far wake of each turbine. The RANS models become more similar further down in the row and the recovery behind the last turbine (aka. the wind farm recovery) is very similar. In van der Laan et al. (2015c), a similar observation was made when comparing the standard $k$–$\varepsilon$ and $k$–$\varepsilon$–$f_P$ models for wind farm cases since the difference between both turbulence models reduces with increased levels of turbulence.

The turbulence intensity contours in Fig. 22 show more pronounced differences between the models, e.g., the LES has peaks of TI in the wake shear layers, whereas the RANS models have TI more evenly distributed over the wake. As was also seen and discussed in the V80 case, there is no induction zone of TI for the LES and the development of TI is delayed. Again also the WJ-EARSM has lower TI compared to the $k$–$\varepsilon$ and $k$–$\varepsilon$–$f_P$ models and is in better agreement with the LES.




In conclusion, the WJ-EARSM appears numerically stable and well-behaved (e.g., no monotonic decreasing velocity deficit or other unphysical effects) for interacting wakes, and as in the single wake case, there are both some improvements and some
less desirable effects of the model over the $k$–$\varepsilon$–$f_P$ model.

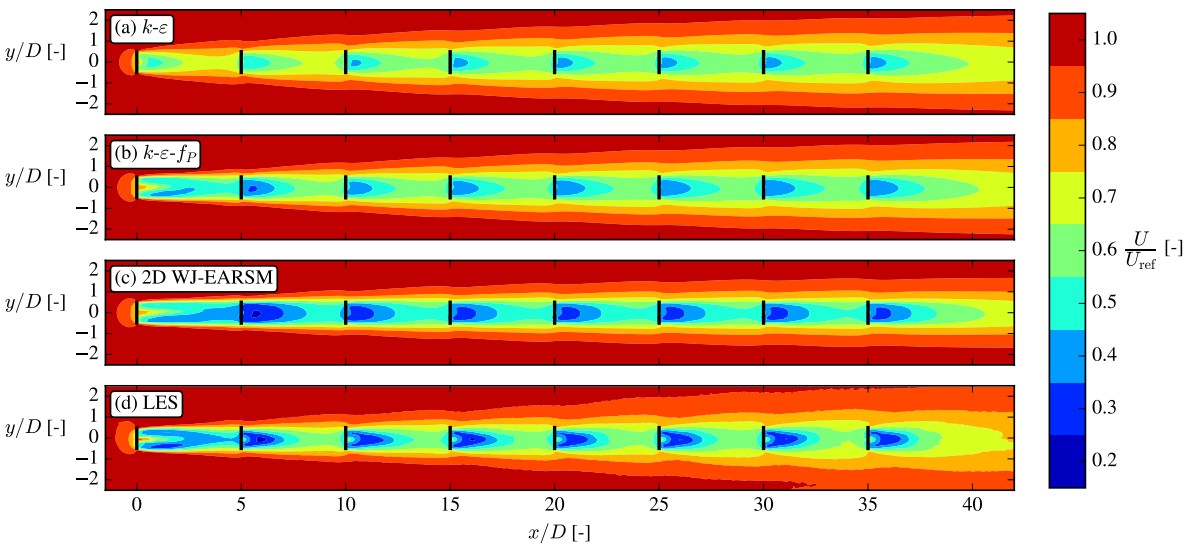

**Figure 21.** Streamwise velocity contour at hub height for the aligned row TotalControl case.

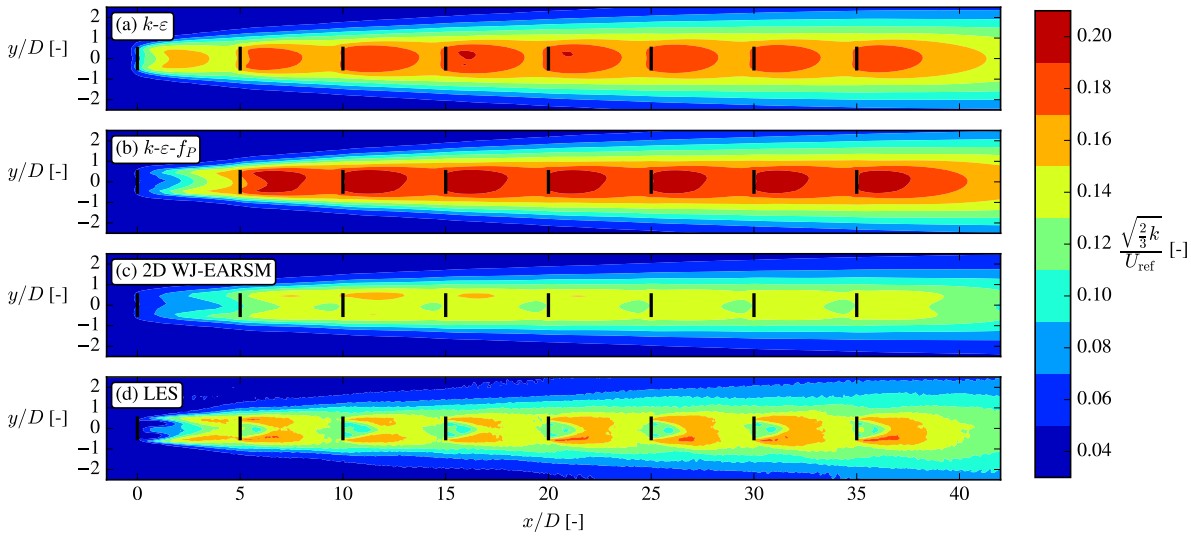

**Figure 22.** TI contour at hub height for the aligned row TotalControl case.





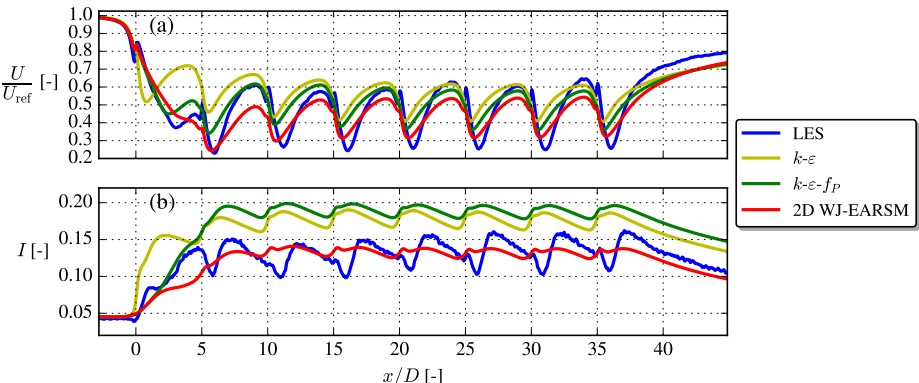

**Figure 23.** Streamwise velocity and turbulence intensity at the axial line going through the AD centers for the aligned row TotalControl case.

## 6 Conclusions

In this paper, we have implemented and applied an EARSM (Wallin and Johansson, 2000) as a turbulence model for RANS simulations of wind turbine wakes in the neutral ASL. To our knowledge, EARSM is rarely, if ever, used in the wind energy community, but we show that it is actually relatively straightforward to implement in CFD codes with already implemented
two-equation turbulence models and importantly that it also appears to be numerically stable for wake simulations. Previous attempts by van der Laan (2014) of applying EARSMs (Taulbee (1992), Gatski and Speziale (1993) and Apsley and Leschziner (1998) models) for wake simulations showed problems with numerical stability, even for single wake cases, but this appears to not be an issue for the EARSM of Wallin and Johansson (2000). The reason for the better numerical behaviour most likely lies in the self-consistent formulation of particular importance in regions with rapid shear, hence preserving physical realizability.

Three canonical flow cases, homogeneous shear flow, half-channel flow and square duct flow, were used to verify the implementation of the model and also showcased some of the advantages with an EARSM over traditional linear EVMs, namely the prediction of freestream turbulence anisotropy and secondary flow phenomena. All three cases have either analytical asymptotes or DNS data to compare against and are easy to setup, which makes them ideal for verification purposes.

For neutral ASL inflow we show that there is a delicate relationship between the turbulence constants that needs to be fulfilled
to ensure a non-developing freestream solution and that it also dictates the amount of freestream turbulence anisotropy. It should be noted that this balance of constants is also important for numerical robustness. Comparing the RANS inflow with reference LES data shows that the WJ-EARSM is capable of predicting similar freestream anisotropy, whereas the turbulence of the $k$–$\varepsilon$–$f_P$ model is almost completely isotropic by definition of its Boussinesq closure. This is also clear from the eigendecomposition of the Reynolds stress tensor, which is visualized with the Barycentric map technique.

A single wake case was first considered and it was observed that the 2D version of the EARSM yielded almost identical results to the 3D version, even when tangential forces were applied on the AD, and we used the 2D model for the remainder of the paper. It should be noted that the 2D version of the WJ-EARSM is a complete and invariant model for general 3D mean





flows. Only the particular dependency of pure 3D effects are simplified, which will have minor effects in most 3D mean flows of interest. The wake profiles of the EARSM model were more top-hat shaped than the profiles observed in the LES data, which might be related to the underlying weak-equilibrium assumption and limitations in the length-scale determining $\varepsilon$ model equation. The disk-averaged velocity deficit and turbulence intensity recovery profiles were however improved over the $k$–$\varepsilon$–$f_P$ model for the specific case. It is possible to obtain a more Gaussian shaped wake profile by increasing the $c_1$ constant and re-tuning the turbulence model constants, but this will deteriorate the prediction of the underlying ASL anisotropy (Appendix B). The wake in real conditions, as well as in LES simulations, will be subjected to slow movements due to large scale turbulence in the ASL and possibly instabilities in the wake development. All such superimposed movements will enforce a more Gaussian-shaped wake, which is demonstrated by a post-processing step in the form of a Gaussian filter (Appendix C). A notable difference between the WJ-EARSM and the $k$–$\varepsilon$–$f_P$ model is also that the latter predicts large peaks of lateral and vertical normal Reynolds stress components in the wake, which are not present in the LES data, because of the Boussinesq closure; this deficiency and the low value of $C_\mu$ used for the $k$–$\varepsilon$–$f_P$ model are possibly the reasons why it tends to overpredict turbulence intensity in the wake.

Finally, we simulated the lower row of the TotalControl reference wind farm consisting of ten aligned turbines, where the trends from the single wake case could also be seen, e.g., the top-hat shaped profiles and better turbulence intensity prediction with EARSM. There were more uncertainties on the comparison with the LES data in this case, because different turbine modeling techniques were used, but the case nevertheless shows that the EARSM also behaves sensible in cases with wake–wake interaction.

In conclusion, the EARSM of Wallin and Johansson (2000) can be used for wake simulations in a numerical robust way and only has a small computational overhead compared to standard two-equation models, hence it is at least three orders of magnitude faster than LES. It provides an advantage over two-equation models in the sense that it has more realistic inflow with anisotropic turbulence and that the wake turbulence also becomes more anisotropic, which indeed is also observed in LES. The turbulence intensity prediction was improved for both test cases considered, while velocity deficit was only considerably improved for the single wake case; more cases (both LES and experimental data) are needed to draw general conclusions about its performance in this regard.

Atmospheric conditions in thermally stable stratification and thermal convection is strongly influencing the turbulence states, anisotropies and, in particular, the vertical mixing in the ASL. This will have a fundamental influence on the wake development and the performance of wind parks. The extension of the EARSM to non-neutral conditions has over the recent years been developed by Lazeroms et al. (2013) and Želi et al. (2019) have demonstrated the model's capability of capturing these effects. This will be of interest for future wind turbine wake studies.



## Appendix A: Grid study of WJ-EARSM for wind turbine wake simulations

Earlier studies (van der Laan et al., 2015b) have shown that a grid spacing of $D/8$ in the wake region is sufficient for grid
convergence of wake velocity deficits with the $k$–$\varepsilon$–$f_P$ model and the same conclusion can be drawn for the 2D WJ-EARSM,
see Fig. A1. We also plot the TI profiles in Fig. A1, which are more sensitive to grid resolution, but we nevertheless decide to
use $D/8$ in this paper, because it represents the typical resolution used in wind farm studies and saves a considerable amount
of computational resources.

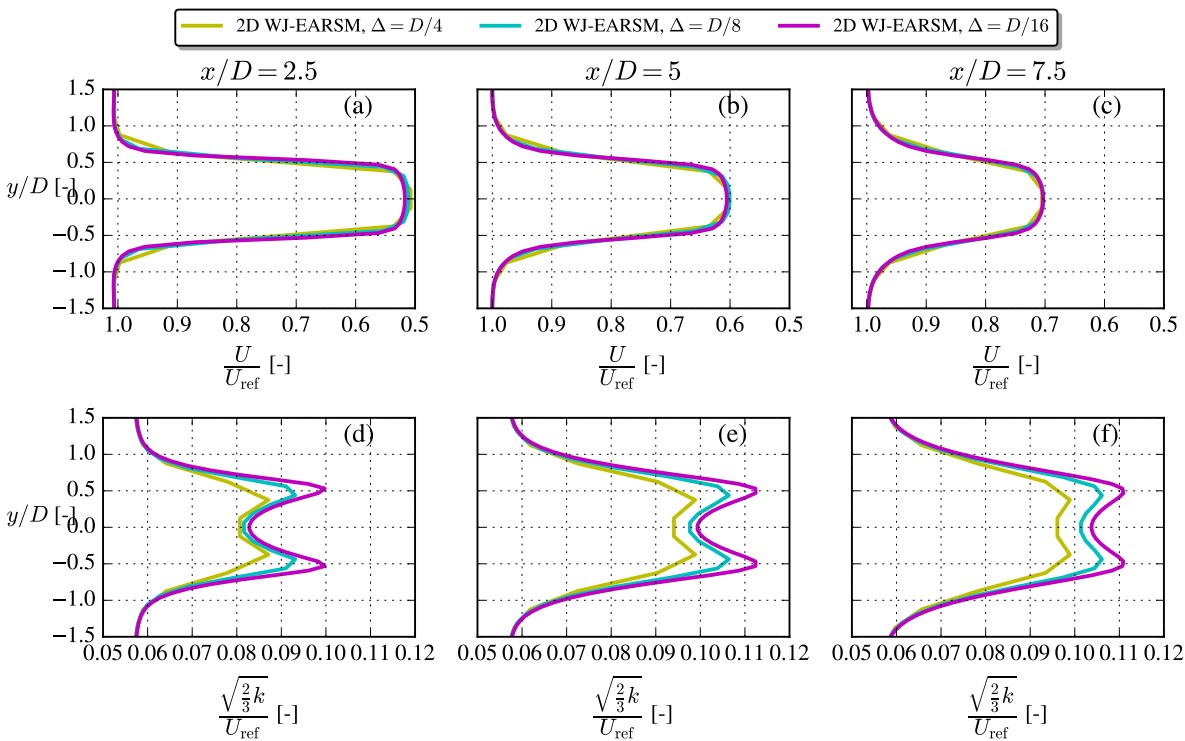

**Figure A1.** Grid study of streamwise velocity (upper row) and TI (lower row) profiles at hub height for the V80 case with the 2D WJ-EARSM
using different mesh resolutions.

## Appendix B: Tuning the turbulence model constants in the WJ-EARSM

The $c_1$ constant, aka. Rotta coefficient, in the WJ-EARSM originates from the pressure-redistribution term and can in principle
be re-tuned, e.g., the original LRR model has $c_1 = 1.5$ and van der Laan (2014) used $c_1 = 4.5$ for the $f_P$ model. As shown in
Sect. 4.1, for the neutral ASL the WJ-EARSM can be used to obtain a direct relationship between $c_1$ and many other variables
in equilibrium, see Fig. 11. From this it is clear that the ASL anisotropy is enhanced for decreasing $c_1$ and vice versa. It is





important to emphasize that the variables in Fig. 11 are *only* dependent on $c_1$ and not the other turbulence constants. In this
section we will test three different $c_1$ values in addition to the standard value of $c_1 = 1.8$, see Table B1.

| Constants | | Derived for general, neutral ASL | | | | | For V80 case | |
|---|---|---|---|---|---|---|---|---|
| Case | $C_{\varepsilon,1}$ | $C_\mu$ | $a_{13}$ | $a_{11}$ | $\sigma_v/\sigma_u$ | $\sigma_w/\sigma_u$ | $z_0$ [m] | $u_*$ [m/s] |
| $c_1 = 1.2$ | 1.44 | 0.085 | -0.29 | 0.37 | 0.80 | 0.53 | $2.93 \cdot 10^{-3}$ | 0.30 |
| $c_1 = 1.8$ | 1.44 | 0.087 | -0.30 | 0.25 | 0.85 | 0.68 | $3.12 \cdot 10^{-3}$ | 0.30 |
| $c_1 = 4.0$ | 1.34 | 0.054 | -0.23 | 0.11 | 0.93 | 0.85 | $8.88 \cdot 10^{-4}$ | 0.27 |
| $c_1 = 8.0$ | 1.18 | 0.030 | -0.17 | 0.06 | 0.96 | 0.92 | $1.50 \cdot 10^{-4}$ | 0.23 |

**Table B1.** Tested sets of turbulence model constants and derived variables for the single wake V80 case. For all sets, we use $C_{\varepsilon,2} = 1.82$,
$\sigma_k = 1.0$, $\sigma_\varepsilon = 1.3$, $\kappa = 0.38$ and $c_2 = \frac{5}{9}$.

Equation (23) needs to be satisfied to have a balanced RANS solution, hence when $c_1$ is adjusted and $C_\mu$ thereby changes,
then either $\kappa$, $\sigma_\varepsilon$, $C_{\varepsilon 1}$, $C_{\varepsilon 2}$ or a combination of all needs to be adjusted. We choose to adjust $C_{\varepsilon 1}$ and fix the others, and the
resulting sets of constants are shown in Table B1. The first set ($c_1 = 1.2$) gives anisotropic freestream turbulence close to the
Panofsky and Dutton (1984) neutral ASL values, the second set ($c_1 = 1.8$) is equivalent to the standard set (see Table 2), the
third set ($c_1 = 4.0$) gives $C_\mu = 0.054$, which is close to the value used by WAsP CFD (Bechmann, 2016) and finally the last
set ($c_1 = 8.0$) gives $C_\mu = 0.03$, which is often used for atmospheric applications (Sørensen, 1995; Richards and Hoxey, 1993).

Another consideration is that the roughness length and friction velocity also needs to be modified to give the same hub
height velocity and turbulence intensity according to Eq. (22), again because $C_\mu$ changes with changing $c_1$. The values of these
are therefore also included in Table B1 and explain why the velocity inflow profiles differ slightly in Fig. B1. The figure also
clearly demonstrates that freestream turbulence anisotropy decreases for increasing $c_1$ and vice versa, which is also evident
from the combination of Eq. (10-11):

$$\frac{\beta_4}{\beta_1} = N^{-1} = \left( \frac{9}{4} c_1 \right)^{-1} . \tag{B1}$$

In the derivation of Eq. B1 we use the definition of $N$, $c_1'$ and that $\mathcal{P}/\varepsilon = 1$ in the neutral ASL. From Eq. (B1), we see that for
increasing $c_1$ there is a decreasing $a_{ij}^{(\mathrm{ex})}$, see Eq. (9), which is the part of the closure responsible for anisotropy and therefore
explains why the turbulence becomes more isotropic with increasing $c_1$.

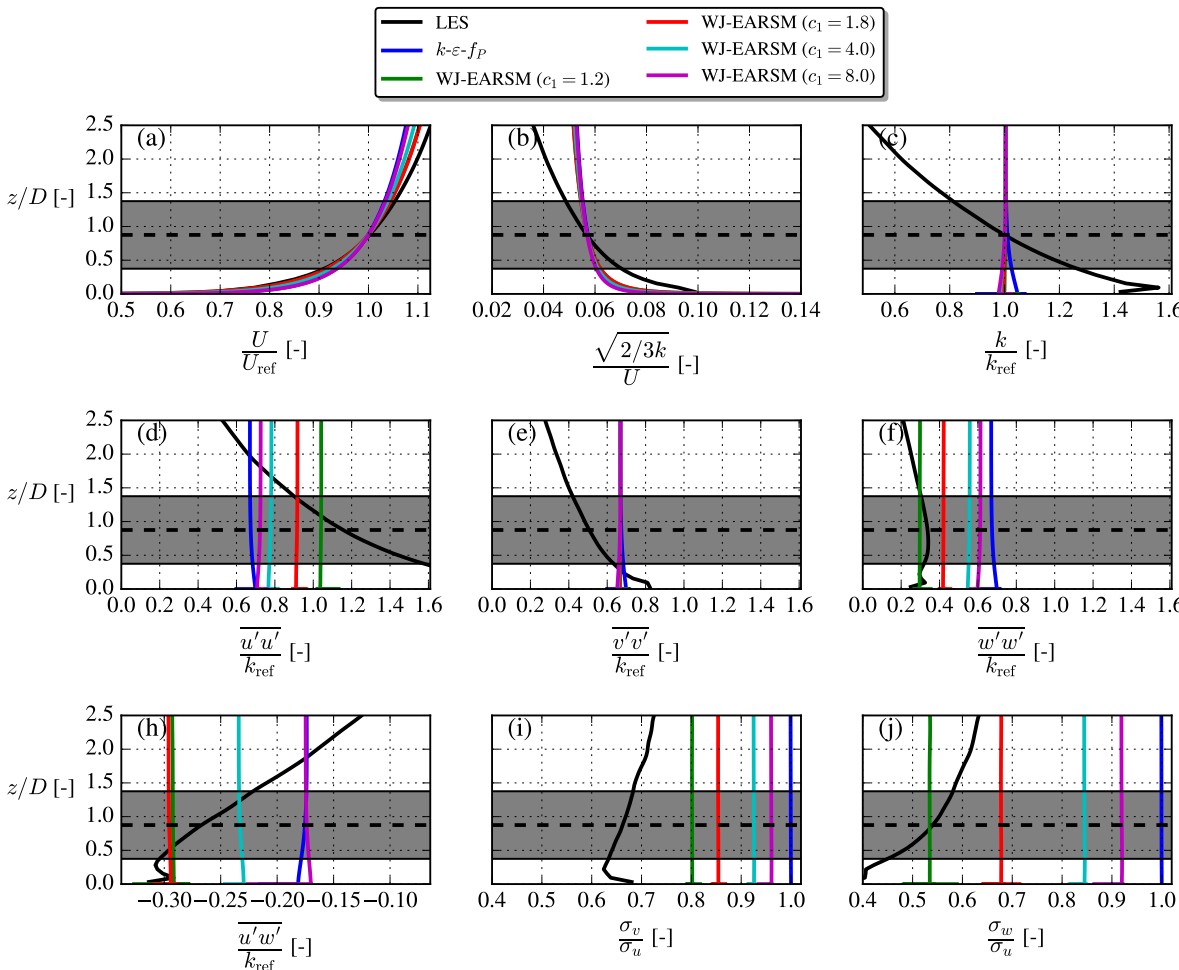

**Figure B1.** Inflow profiles for the single wake V80 case with different sets of model constants.

Fig. B2 shows how the wake is effected by the new sets of constants. The velocity deficit shape is more Gaussian for larger $c_1$ and thus more similar to the LES shape, while the turbulence intensity increases. From these observations, one could argue that $c_1 = 4.0$ would perhaps be a better choice for modeling of velocity deficit, while $c_1 = 1.8$ is better for TI and anisotropy predictions.

Increasing $c_1$ also has a significant impact on the normal stress contours as shown in Fig. B3, where one especially can note how the length of the inner low $\overline{u'u'}$ core decreases. In this regard a larger $c_1$ than the standard $c_1 = 1.8$ also seems desirable, although we again have to remind that this will also result in less correct freestream anisotropy. However, it it notable that although a larger $c_1$ leads to more isotropic freestream turbulence, then in contrast to the $k-\varepsilon-f_P$ model (see Fig. 17), there is still significant wake turbulence anisotropy.



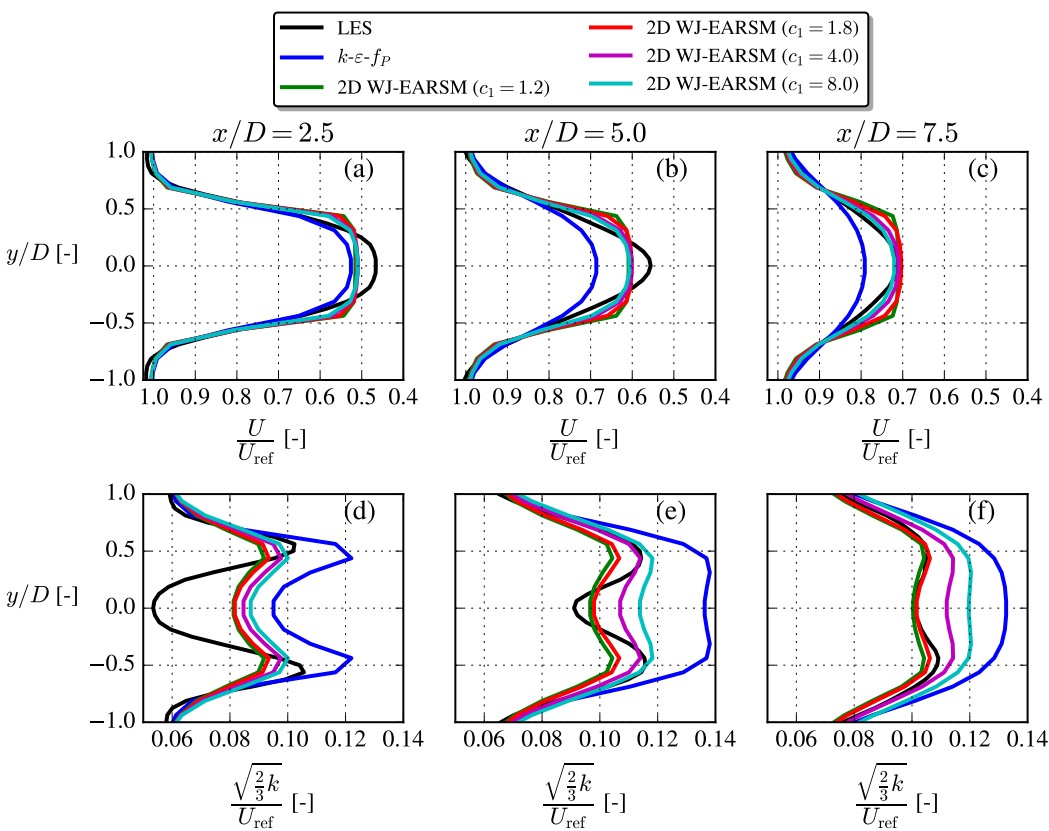

**Figure B2.** Streamwise velocity (upper row) and TI (lower row) profiles extracted at various donwstream positions for the single wake V80 case with different sets of model constants.

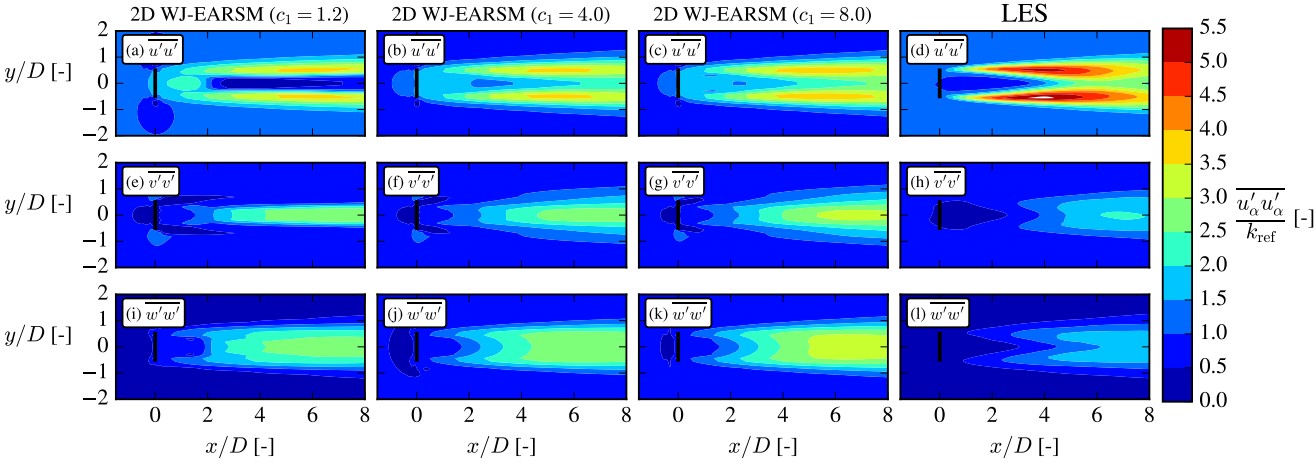

**Figure B3.** Normal stresses at hub height for the single wake V80 case with different sets of model constants.





## Appendix C: Gaussian filtered RANS model

As discussed in Sect. 4.2 and Appendix B, the WJ-EARSM velocity deficit is rather top-hat shaped, whereas the LES shape is more Gaussian. This could also be interpreted as a consequence of unaccounted large scale atmospheric turbulence, wake instabilities and the applied unidirectional inflow in our RANS simulation. In this section we present a possible post-processing step to take this into account.

Defining the wind direction as $\varphi \equiv \arctan\left(\frac{V}{U}\right) \approx \frac{V}{U}$ (small angle approximation), we can calculate its variance as:

$$\sigma_\varphi^2 = \left(\frac{\partial \varphi}{\partial U}\right)^2 \sigma_u^2 + \left(\frac{\partial \varphi}{\partial V}\right)^2 \sigma_v^2 = \left(-\frac{V}{U^2}\right)^2 \sigma_u^2 + \left(\frac{1}{U}\right)^2 \sigma_v^2 \approx \frac{\sigma_v^2}{U^2}. \tag{C1}$$

Taking the hub height LES values for the V80 case gives $\sigma_\varphi = \pm 0.0484 \text{ rad} = \pm 2.8°$. Assuming zero mean wind direction and a Gaussian distribution gives a simple model for the wind direction variability:

$$f_\varphi(\varphi) = \frac{1}{\sigma_\varphi \sqrt{2\pi}} e^{-\frac{1}{2}\left(\frac{\varphi}{\sigma_\varphi}\right)^2}. \tag{C2}$$

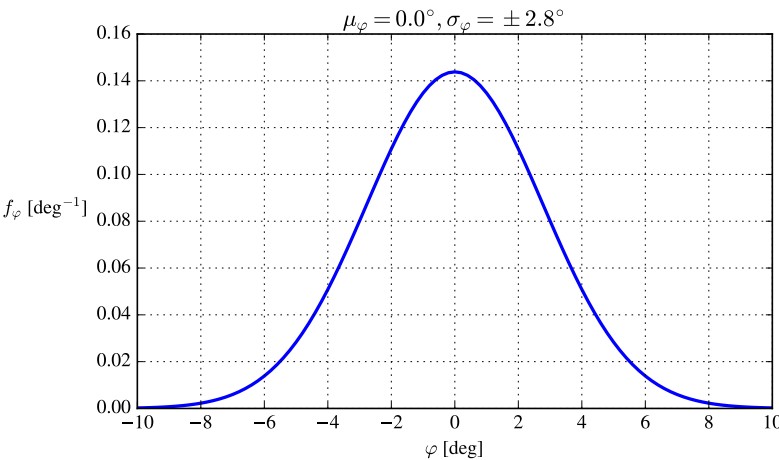

**Figure C1.** Gaussian wind direction distribution.

Perhaps the simplest way to model the effect of wind direction variability in RANS is to just run a single simulation with wind direction equal to zero (as we usually do) and then apply a Gaussian filter to the wake profiles as a post-processing step. To filter the $U(y)$ profiles from Fig. 15, one needs to convert the wind direction uncertainty to the corresponding uncertainty in the $y$-direction: for small angles, we have $\sigma_y(x) \approx x\sigma_\varphi$, so our Gaussian distribution is then:

$$f_y(y;x) = \frac{1}{\sigma_y \sqrt{2\pi}} e^{-\frac{1}{2}\left(\frac{y}{\sigma_y}\right)^2}. \tag{C3}$$

The distribution is dependent on the downstream position, $x$, and from Fig. C2 one can observe that even a seemingly small $\sigma_\varphi$ leads to a large $\sigma_y$, when one is far downstream. For reference, a standard deviation of $\sigma_\varphi = 5°$ is also shown.





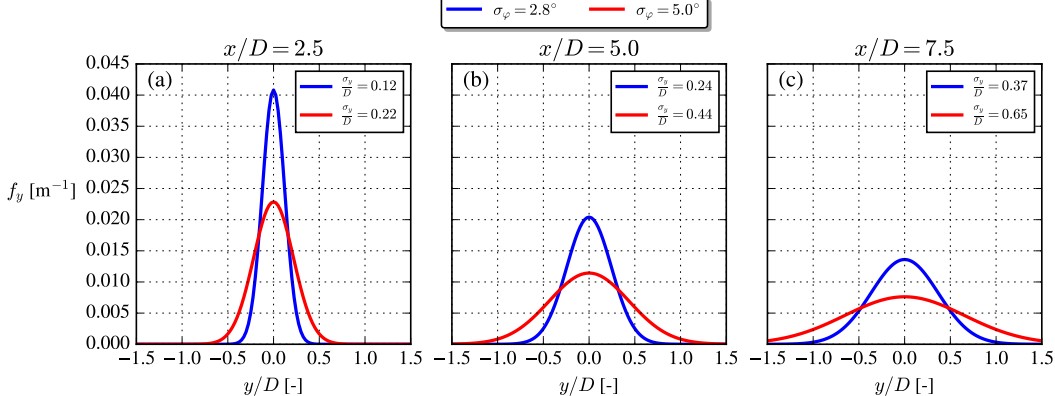

**Figure C2.** Wake profile uncertainty at three downstream distances modeled after Gaussian distributions.

The distributions in Fig. C2 are used for rolling averaging of the 2D WJ-EARSM wake profiles and the results are shown in Fig. C3. A significant effect is seen for both the velocity deficit and TI, and the small uncertainty of $\sigma_\varphi = 2.8°$ calculated with Eq. (C1) is enough to smoothen the sharp edges of the velocity deficit profiles. However, the TI results are not improved by the 595 Gaussian filter.

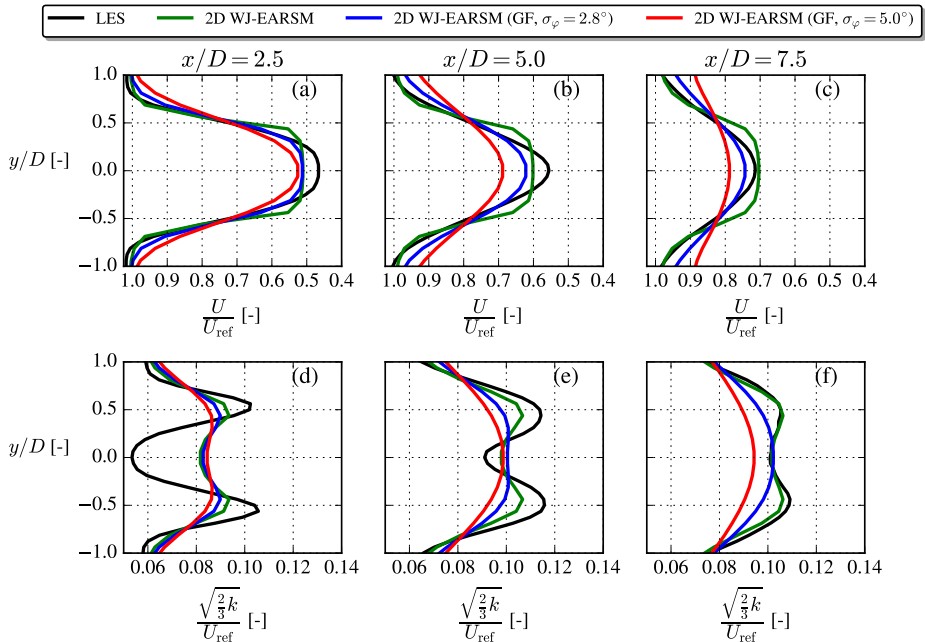

**Figure C3.** Gaussian filtered (GF) wake profiles.



*Code availability.* EllipSys3D is proprietary software of DTU.

*Data availability.* The RANS results were generated with EllipSys3D, but the data presented can be made available by contacting the corresponding author. Interested parties are also welcome to hand-digitize the results and use them as reference in other publications.

*Author contributions.* MB implemented the EARSM in EllipSys1D/3D, performed the RANS simulations and wrote the initial article draft.
600 SW provided guidance for the implementation and verification of the EARSM, MPvdL suggested re-tuning the model constants and using a Gaussian filter to compensate for the top-hat shaped wake profiles, while MK contributed with the discussions regarding inflow turbulence in the neutral ASL. All authors (MB, SW, MPvdL and MK) contributed with editing and finalizing the paper.

*Competing interests.* The authors declare that they have no conflict of interest.

*Acknowledgements.* The authors would like to thank Mahdi Abkar for sharing the LES data of the V80 case and Ishaan Sood and Johan
605 Meyers for making the TotalControl LES data publicly available in an online repository.



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
