# Peer review of "Wind turbine wake simulation with explicit algebraic Reynolds stress modeling"

_Wind Energy Science, 2022_

## Referee Comment (RC2)

**Referee's comments to wes-2022-50**

*Stefano Letizia*

**General comments**

This paper presents a thorough assessment of a non-linear eddy viscosity model for RANS wind farm simulations. The scope is well-defined and very relevant in the context of CFD models for wind farm flows since there are significant flaws in traditional two-equation turbulence closures with linear eddy viscosity to model wakes.

The literature is fairly reviewed and adequate credit is given to previous research work done on this topic.

The model's equations are described with a great level of detail and seem accurate. A possible source of confusion for the reader may be the terminology used to describe the 2D and 3D EARSM which seems to overlap with the 1D and 3D flow simulations. It should be clarified in Section 2 that both 2D and 3D turbulent closures can be applied to three-dimensional turbulence.

The validation through the three idealized flow cases is quite interesting and the reviewer appreciated the effort devoted to this section, which is useful to strengthen the arguments of the paper.

The results are discussed thoroughly and with commendable intellectual honesty even in the cases where there is not a clear improvement provided by the new model.

The paper shall be accepted with minor revisions according to the reviewer, as indicated in the main and specific comments provided below.

**Main comments:**

- Explicitly state the additional computational time compared to the other models, even just for the single wake case.
- Provide the boundary conditions used for the wind turbine row simulations in Section 5. This is a tricky setup since the boundaries of the domain are very close to the rotors and no symmetry seems to apply at the upper boundary due to the staggered layout.
- Remove Appendix B; although it may be true that the LES may be able to capture better the wake meandering induced by the larger incoming eddies, the Gaussian convolution method seems weak since it includes all the lateral turbulent fluctuations as wind direction variations. A well-known rule of thumb states that just eddies larger than 2 rotor diameters induce wake meandering, so using $\sigma_v^2$ as the total "meandering energy" seems an overestimation. Furthermore, the wake is moved as a rigid body according to equation C3, which would imply a perfectly frozen turbulence, which is not realistic since smaller eddies decay way faster than the full wake length. Considering the limited improvement seen in Fig. C3 and the fundamental questions that this appendix may arise, it is suggested to remove it and mention the possible lack of wake meandering as a cause of the top-hat profiles.

**Specific comments**

Line 4: please use "alleviate" instead of "aid" when referring to deficiencies.

Lines 29-40, "Linear…applications": please add a reference to support this statement.

Line 56: Please consider "is chosen in this work" instead of "preferred" to limit the generality of this statement.

Lines 86-87, "This paper…flows": please add a reference to support this statement.

Lines 97-00, "The more…$\Omega_{ij}$": this statement is not clear since also the standard model solved the transport equations of $\kappa$ and $\epsilon$.

Line 124: Please define "RDT".

Line 200: was "abandoned" intended instead of "abounded"?

Line 201: the formulation of the production cannot be derived readily, please provide derivation (in appendix) or reference.

Lines 246-249, "In principle…code": this statement is unclear and unnecessary. Please clarify or remove.

Line 300: The fact that the log law profiles hold for $z < 0.3 L_z$ agrees with [Pope S., Turbulent Flows, 2000], which could be cited.

Lines 361-376: At the end of this discussion, it is not clear which values are utilized for the simulations. Please state the value of the turning constants clearly.

Line 504: the Boussinesq's closure leads to a Reynolds stress tensor that is inherently isotropic only for the diagonal terms $\overline{u_i u_i} = \frac{2}{3} k$, but not for the off-diagonal (shear stress) terms, which can produce different eigenvalues. It is recommended to remove "by definition of the Boussinesq's closure".

Line 554: Please clarify "behave sensible".

Appendix seems not to be cited in the manuscript.

Line 591: Does the expression "rolling average" mean that the RASN profiles are "convoluted" using the pdf of the wake centers in Fig. C2?

---

## Author Comment (AC1)

**Reply to reviewers**

**August 10, 2022**

We would like to thank the two reviewers for undertaking the job of reviewing our work. The suggestions are much appreciated and have contributed to the value as well as the clarity of the paper. Below you find copies of the reviewers comments and our answers (blue color), while an additional document is provided that highlights all modifications of the paper with respect to the initial submitted version.

**Reviewer 1**

This paper discusses the use of an Algebraic Reynolds Stress Model for the simulation of wind turbine wakes. The paper validates the model for some simpler test cases before considering the wake and multiple wake cases. The paper is well written and technically correct. The turbulence model being considered is actually fairly simple, but every application goes through this sort of evolution to more complex turbulence models, and it is useful for the applications to know what the turbulence model cost and accuracy trade-offs are for their particular application.

**Specific comments:**

Eq. (7):

1. Line 86: This a somewhat odd description of k-omega vs k-epslon. The SST k-omega model is a fixed-up version that is a blend of k-epsilon and k-omega and that gets most of the advantages of each.

We agree and have changed the sentence to "This paper focuses on the  $k-\varepsilon$  model, because it is traditionally used for atmospheric flows (e.g., Crespo et al., 1985; Richards and Hoxey, 1993; Sørensen, 1995), but the widely used  $k-\omega$  model could as well have been used."

2. Line 119: "In the freestream of a neutral ASL, fP = 1,": It looks like it = 2\*f0, not 1. It is actually equal to 1. Proof: In the freestream, we have  $\sigma/\tilde{\sigma} = 1$  by definition. Inserting this into

$$f_P = \frac{2f_0}{1 + \sqrt{1 + 4f_0(f_0 - 1)(\sigma/\tilde{\sigma})^2}} \tag{1}$$

$$=\frac{2f_0}{1+\sqrt{1+4f_0(f_0-1)}}\tag{2}$$

$$=\frac{2f_0}{1+\sqrt{(2f_0-1)^2}}\tag{3}$$

$$=1$$
 (4)

3. Table 1: The Ce2 value is the "old fashioned" and too high value. The Ce1 is far too small (should be close to 1.5 to get lengthscales right), and this is probably why  $C_{\mu}$  is far too small. When this model fails later is it because of the linear eddy viscosity assumption, or because of the poor model constants? Table 2 has much better (more modern) model constants.

For atmospheric applications there is indeed some controversy about the choice of either the "far too small"  $C_{\mu} = 0.03$  (Table 1) or the more accepted value of the turbulence community  $C_{\mu} = 0.09$  (Table 2), see discussions by Bottema (1997) and Richards and Norris (2011), which were also cited just before Table 1. The k- $\varepsilon$ - $f_P$  model was calibrated by van der Laan (2014) with the model constants from Table 1, which is why these were also used in the current paper. In Fig. 14 and 15, we also simulated the single wake case using the standard k- $\varepsilon$  model with the "modern" model constants from Table 2, but it only gave a marginal improvement over the standard k- $\varepsilon$  model with the model constants from Table 1. Also, no matter the choice of model constants the linear EVMs will never be able to fully predict anisotropy or secondary motions.

4. Line 160: "ensures good model predictions in nonequilibrium conditions": Equilibrium means different things to different turbulence modelers. All ARSM are assuming a form of equilibrium. They are assuming that the turbulence anisotropy instantly responds to the mean flow (is in equilibrium with the mean flow). Non-equilibrium is captured by a full RST model (solving a transport model for the Reynolds stress anisotropy with a time derivative and an advection term). An example of non-equilibrium turbulence (for many modelers) is a stagnation point or boundary layer separation point (on an airfoil). Where the strain is zero, and the turbulence anisotropy is non-zero (because it is advected or diffused there).

We have clarified the meaning of "non-equilibrium" in Line 160 by adding "for  $\mathcal{P}/\varepsilon > 1$ ". Also in the introduction we have now added "non-equilibrium conditions (where TKE production does not balance dissipation of TKE)".

5. Line 200: "was later abounded for" change to abandoned

Yes, thanks for catching this typo.

6. Section 2.4 (or a new section) maybe should discuss the boundary conditions. The k-epsilon system presented doesn't work properly near a solid surface. Also how are k and epsilon specified at inlets? K can never be zero - or the model fails.

References to Sørensen et al. (2007); Sørensen (1995) have been added to Section 2.4, which can be consulted for details on the numerical implementation/formulation of the different types of boundary conditions in EllipSys3D. Indeed the standard k- $\varepsilon$  model fails, when integrating all the way to the wall, which is why we use the aerodynamic rough wall (Sørensen et al., 2007). The inlet BC is only used for the wake simulations, where k and  $\varepsilon$  are set using the log-law profiles from Eq. (22), see Section 4.1.

7. Section 3.1: This test case reveals very little. The constants in the models were explicitly chosen to fit this test case.

There is a distinction between "validation" (the process of assessing a model's accuracy) and "verification" (the process of ensuring the correct implementation of the model), see Réthoré et al. (2014). Both reviewers have used "validation" for describing Section 3, but it is primarily intended as a verification section, hence the section name "verification cases". Indeed in the context of validation the homogeneous shear flow case reveals little, however in the context of verification it is a valuable case to confirm the correct implementation of the EARSM. This distinction has now been clarified at the beginning of Section 3.

8. Table 3 can be deceptive. The goal of these models is to predict the mean flow well, not the anisotropy. Only the accuracy of the divergence of the anisotropy really matters. LES models, function similarly. They predict the unresolved stresses terribly, but the resolved flow reasonably (because their divergence is roughly correct). Figure 4 confirms this understanding. Only a13 really matters and the most recent constants have been tuned to get that stress correct at the expense of the stresses that don't matter.

Indeed we also realize during this study that if the goal is to predict the correct  $U_i$  field, the shear stress and its divergence are the most important components - although this might be well known in the turbulence community, we believe that it is less known in the wind energy community and have therefore quantified it further in a follow-up brief communication, which is also currently in discussion (https://doi.org/10.5194/wes-2022-56). If a secondary goal of the model is to also be able to predict anisotropy, second order statistics (for example TI) and secondary motions, then it is also important to be able to predict the other components of  $a_{ij}$ . If a more general diffusion model is used

(not used in the current study), e.g., the Daly-Harlow model, then the vertical normal component,  $a_{33}$ , is also important, because it enters directly in the diffusion terms of the k and  $\varepsilon$  equations, see Wallin and Johansson (2000). We believe that Table 3 is still useful for verification purposes, see answer to 7.

**General comments**

The Gaussian shape of the axisymmetric turbulent wake can be obtained by using similarity theory and assuming an eddy viscosity that is constant across the wake (but varies in the streamwise direction). This knowledge can provide a strong hint as to why a model works in a wake and why it doesn't (does it produce near-constant eddy viscosity). The ARSM results are quite bad for the wake velocity given that a Prandtl mixing length model (eddy-viscosity = constant \* wake width \* velocity deficit) works better.

There is an old axisymmetric vs planar jet modeling problem (Durbin text book probably has it). Few models can predict both cases well. Is it possible you have an ARSM tuned for planar shear flows that now struggles in the axisymmetric wakes you actually want to model.

The derivation of the self-similar axisymmetric turbulent wake assumes (i)  $x/D \gg 1$  (so that the geometry of the object is "forgotten" and the thin-shear layer equation can be used) (ii) that no wall is present (so that inflow is uniform) and (iii) that the eddy viscosity is constant in the wake. While the self-similar model is very elegant, the assumptions stated above limit its range of applicability, which motivates the use of the less elegant, but more general CFD methods.

It is an interesting analogy and whether an equivalent problem exist with axisymmetric vs. plane wake is unbenownst to us; the authors have not simulated jet flows with the EARSM, but other axisymmetric problems has been investigated with success, e.g., the rotating pipe flow (Wallin and Johansson, 2000) and the decaying vortex Wallin (2000).

**Reviewer 2**

**General comments**

This paper presents a thorough assessment of a non-linear eddy viscosity model for RANS wind farm simulations. The scope is well-defined and very relevant in the context of CFD models for wind farm flows since there are significant flaws in traditional two-equation turbulence closures with linear eddy viscosity to model wakes. The literature is fairly reviewed and adequate credit is given to previous research work done on this topic. The model's equations are described with a great level of detail and seem accurate. A possible source of confusion for the reader may be the terminology used to describe the 2D and 3D EARSM which seems to overlap with the 1D and 3D flow simulations. It should be clarified in Section 2 that both 2D and 3D turbulent closures can be applied to three-dimensional turbulence. The validation through the three idealized flow cases is quite interesting and the reviewer appreciated the effort devoted to this section, which is useful to strengthen the arguments of the paper. The results are discussed thoroughly and with commendable intellectual honesty even in the cases where there is not a clear improvement provided by the new model. The paper shall be accepted with minor revisions according to the reviewer, as indicated in the main and specific comments provided below.

We acknowledge that the terminology can be confusing, but chose to use it to be consistent with previous litterature (Hellsten and Wallin, 2009; Lazeroms et al., 2013; Wallin and Johansson, 2000; Želi et al., 2019). In Section 2.3 (just before presenting the 2D and 3D models), we already tried to clear up this possible confusion: "It is important to stress that the 2D model is fully general and invariant and can be used for simulation of 3D mean flows as noted by Hellsten and Wallin (2009); this has for example also been commonly done with the EARSM of Gatski and Speziale (1993)".

**Main comments**

• Explicitly state the additional computational time compared to the other models, even just for the single wake case.

Adopted. The computational overhead of the EARSM was found to be negligible compared to the k- $\varepsilon$ /k- $\varepsilon$ - $f_P$  models and EARSM was even found to converge faster for some cases. For the single wake case the timings of all the simulations (except the LES) shown in Fig. 14 were for example within  $\pm 5\%$  of each other; this has been added to the conclusion. However, it should be said that each model was only run once, which could lead to some error of the estimate, since the code was run on a HPC, where timings sometimes vary for the exactly same simulation. Also the scaling of the model has not been investigated. The focus of the current paper has more been on "proof-of-concept" rather than numerical efficiency.

• Provide the boundary conditions used for the wind turbine row simulations in Section 5. This is a tricky setup since the boundaries of the domain are very close to the rotors and no symmetry seems to apply at the upper boundary due to the staggered layout.

Adopted. We realize that the setup for the aligned row case was not described sufficiently and that Fig. 20 might confuse the reader: We did *not* simulate the full staggered wind farm, but only a row of 8 turbines. Also we extended the domain, so that the boundaries were very far from the rotors. The top BC is an inlet, *not* a symmetry BC. All of these points have been clarified and we have chosen to show the domain of our aligned row RANS simulation instead of the TotalControl layout in Fig. 20 to avoid further confusion.

• Remove Appendix B; although it may be true that the LES may be able to capture better the wake meandering induced by the larger incoming eddies, the Gaussian convolution method seems weak since it includes all the lateral turbulent fluctuations as wind direction variations. A well-known rule of thumb states that just eddies larger than 2 rotor diameters induce wake meandering, so using  $\sigma_v^2$  as the total "meandering energy" seems an overestimation. Furthermore, the wake is moved as a rigid body according to equation C3, which would imply a perfectly frozen turbulence, which is not realistic since smaller eddies decay way faster than the full wake length. Considering the limited improvement

seen in Fig. C3 and the fundamental questions that this appendix may arise, it is suggested to remove it and mention the possible lack of wake meandering as a cause of the top-hat profiles.

From the above description, it seems the reviewer means "Appendix C" not "Appendix B". Very useful comments, which has led us to re-write Appendix C completely and to emphasize that it is just an qualitative analysis to estimate the order of magnitude of the effect of wind direction variations. With regard to the "meandering energy", we expect that using the whole of  $\sigma_y^2$  only gives a small overprediction, because most of the variance comes from the large eddies, which as the reviewer points out are the ones responsible for meandering. A larger overprediction of the time-averaged meandering could be expected from the second point made by the reviewer (the assumption of "frozen turbulence" or "rigid wake movement" implied by eq. C3). To counteract this, we have now also included a model with quadratic filter width development, which indeed improves the results as compared with LES. In summary, Appendix C shows in a simple manner that the lack of meandering can explain the tophat shaped wake profile of a steady RANS model, hence its relevance and why we believe it is worth keeping.

**Specific comments**

- 1. Line 4: please use "alleviate" instead of "aid" when referring to deficiencies. Adopted.
- 2. Lines 39-40, "Linear...applications": please add a reference to support this statement. Have added a few recent papers as examples of the statement.
- 3. Line 56: Please consider "is chosen in this work" instead of "preferred" to limit the generality of this statement.

Adopted.

- 4. Lines 86-87, "This paper...flows": please add a reference to support this statement. Per reviewer 1's comment, this sentence has been changed and we have at the same time included some references.
- 5. Lines 97-100, "The more... $\Omega_{ij}$ ": this statement is not clear since also the standard model solved the transport equations of k and  $\varepsilon$ .

The EARSM depends on  $S_{ij} \equiv \frac{1}{2} \frac{k}{\varepsilon} \left( \frac{\partial U_i}{\partial x_j} + \frac{\partial U_j}{\partial x_i} \right)$ , hence we still need to solve the k and  $\varepsilon$  transport equations when using EARSM to obtain  $k/\varepsilon$ . The sentence has been extended to clarify this.

- 6. Line 124: Please define "RDT". Adopted as "rapid distortion theory (RDT)". It refers to the situation where  $S_{\varepsilon}^{k} \gg 1$ .
- 7. Line 200: was "abandoned" intended instead of "abounded"? Yes, indeed this typo was also pointed out by reviewer 1.
- 8. Line 201: the formulation of the production cannot be derived readily, please provide derivation (in appendix) or reference.

The expression is found frequently in turbulence litterature, e.g., (Girimaji, 1996; Taulbee, 1992; Wallin and Johansson, 2000). It is valid for all turbulence models and is derived as:

$$\frac{\mathcal{P}}{\varepsilon} = \frac{-\overline{u_i'u_j'}\frac{\partial U_i}{\partial x_j}}{\varepsilon} \tag{5}$$

$$= \frac{-\left(a_{ij}k + \frac{2}{3}k\delta_{ij}\right)\frac{\varepsilon}{k}(S_{ij} + \Omega_{ij})}{\varepsilon}$$
(6)

$$= -a_{ij}(S_{ij} + \Omega_{ij}) \tag{7}$$

$$= -a_{ij}S_{ij} \tag{8}$$

-

The third equality is because  $S_{ij}$  and  $\Omega_{ij}$  are traceless. The fourth equality is because  $a_{ij}$  is symmetric and  $\Omega_{ij}$  is anti-symmetric (the doubly contraction of such two tensors is zero, see Pope (2000) appendix B). The sentence has been extended with a reference.

9. Lines 246-249, "In principle...code": this statement is unclear and unnecessary. Please clarify or remove.

We believe that the sentence is necessary, because it explains why we use a finite-volume grid (Fig. 4), which might appear strange for readers familiar with homogeneous shear flow - they would most probably *not* use finite-volume: There is no space dependence for homogeneous shear flow (it has zero space dimensions, hence "0D", and only depends on time), so the system of PDEs transform into a set of ODEs, which could be integrated forward in time without space discretization. For example Taulbee (1992) follows the latter approach for his homogenous shear flow simulation and a reference has been added to this.

10. Line 300: The fact that the log law profiles hold for  $z < 0.3L_z$  agrees with [Pope S., Turbulent Flows, 2000], which could be cited.

Adopted.

11. Lines 361-376: At the end of this discussion, it is not clear which values are utilized for the simulations. Please state the value of the tuning constants clearly.

Adopted.

12. Line 504: the Boussinesq's closure leads to a Reynolds stress tensor that is inherently isotropic only for the 2 diagonal terms  $\overline{u_i u_i} = \frac{2}{3}k$ , but not for the off-diagonal (shear stress) terms, which can produce different 3 eigenvalues. It is recommended to remove "by definition of the Boussinesq's closure".

We wrote "almost completely isotropic" instead of "isotropic" for this very reason and it can for example also be seen in Fig. 13 that the Boussinesq hypothesis does not give perfectly isotropic freestream turbulence (although it is quite close to  $x_{3c}$ ). Indeed the three eigenvalues of  $a_{ij}$  in the freestream with the Boussinesq hypothesis are  $\sqrt{C_{\mu}}$ , 0 and  $-\sqrt{C_{\mu}}$  (to be exactly isotropic all eigenvalues of  $a_{ij}$  should equal 0). We have reformulated the sentence to emphasize that it is not exactly isotropic.

13. Line 554: Please clarify "behave sensible".

Adopted.

14. Appendix seems not to be cited in the manuscript.

We presume that the reviewer refers to Appendix A. The appendix is now cited in the beginning of section 4.

15. Line 591: Does the expression "rolling average" mean that the RANS profiles are "convoluted" using the pdf of the wake centers in Fig. C2?

Yes, we now use "convolution" instead and define convolution with an equation. The whole of Appendix C has been completely re-written, see main comment 3.

**Own** improvements**

- Several sentences (mostly in the introduction and conclusion) have been slightly reformulated for either clarification or more correct english usage. See the difference document for these changes.
- Added to the discussion regarding  $C_{\mu}$  that  $C_{\mu} = 0.052$  is used by WAsP-CFD, which is a code commonly used by the industry for wind resource estimation (https://www.wasp.dk/waspcfd).
- It was mentioned several places that a row of ten turbines was simulated, while it was really only eight turbines.

[revised manuscript text omitted]